# Gradient Descent Provably Solves Nonlinear Tomographic Reconstruction

## Abstract

In computed tomography (CT), the forward model consists of a linear Radon transform followed by an exponential nonlinearity based on the attenuation of light according to the Beer–Lambert Law. Conventional reconstruction often involves inverting this nonlinearity as a preprocessing step and then solving a convex inverse problem. However, this nonlinear measurement preprocessing required to use the Radon transform is poorly conditioned in the vicinity of high-density materials, such as metal. This preprocessing makes CT reconstruction methods numerically sensitive and susceptible to artifacts near high-density regions. In this paper, we study a technique where the signal is directly reconstructed from raw measurements through the nonlinear forward model. Though this optimization is nonconvex, we show that gradient descent provably converges to the global optimum at a geometric rate, perfectly reconstructing the underlying signal with a near minimal number of random measurements. We also prove similar results in the under-determined setting where the number of measurements is significantly smaller than the dimension of the signal. This is achieved by enforcing prior structural information about the signal through constraints on the optimization variables. We illustrate the benefits of direct nonlinear CT reconstruction with cone-beam CT experiments on synthetic and real 3D volumes. We show that this approach reduces metal artifacts compared to a commercial reconstruction of a human skull with metal dental crowns.

## 1 Introduction

Computed tomography (CT) is a core imaging modality in modern medicine (Food & Administration, 2023). X-ray CT is used to diagnose a wide array of conditions, plan treatments such as surgery or chemotherapy, and monitor their effectiveness over time. It can image any part of the body, and is widely performed as an outpatient imaging procedure.

CT systems work by rotating an X-ray source and detector around the patient, measuring how much of the emitted X-ray intensity reaches the detector at each angle. Because different tissues absorb X-rays at different rates, each of these measurements records a projection of the patient's internal anatomy along the exposure angle. Algorithms then combine these projection measurements at different angles to recover a 2D or 3D image of the patient. This image is then interpreted by a medical professional (*e.g.* physician, radiologist, or medical physicist) to help diagnose, monitor, or plan treatment for a disease or injury.

CT scanners in use today typically consider the image reconstruction task as a linear inverse problem, in which the measurements are linear projections of the signal at known angles. Omitting measurement noise, we can write this standard linear measurement model as:

$$\hat{y}_i = \boldsymbol{a}_i^T \boldsymbol{x}, \qquad (1.1)$$

where $\boldsymbol{x}$ is a vectorized version of the unknown signal (which commonly lies in 2D or 3D) and $\boldsymbol{a}_i$ is a known, nonnegative measurement vector that denotes the weight each entry in the signal contributes to the integral $\hat{y}_i$ along measurement ray $i$. Computed over a set of regularly-spaced ray angles, this is exactly the Radon transform (Radon, 1917). This linear measurement model is quite convenient, as it enables efficient computations using the Fourier slice theorem—which equates linear projections in real-space to evaluation of slices through Fourier space—as well as strong recovery guarantees from

compressive sensing (Kak & Slaney, 2001; Foucart & Rauhuti, 2013; Bracewell, 1990). This linear projection model is accurate for signals of low density, for which the incident X-rays pass through largely unperturbed.

However, consider the common setting in which the signal contains regions of density high enough to occlude X-rays, such as the metal implants used in dental crowns and artificial joints. Such high-density regions produce nonlinear measurements for which the Fourier slice theorem, and standard compressed sensing results, no longer hold. In practice, tomographic reconstruction algorithms that assume a linear projection as the measurement model produce streak-like artifacts around high-density regions, potentially obscuring otherwise measurable and meaningful signal.

To avoid such artifacts, in this paper we consider a nonlinear measurement model, which correctly models signals with arbitrary density. Equation (1.1) then becomes:

$$y_i = 1 - \exp(-\boldsymbol{a}_i^T \boldsymbol{x}), \tag{1.2}$$

where the exponential nonlinearity accounts for occlusion and is due to the Beer-Lambert Law. In practice, the partial occlusions captured by eq. (1.2) are commonly incorporated into a linear model by inverting the nonlinearity, converting raw measurements $y_i$ from eq. (1.2) into processed measurements $\hat{y}_i = -\ln(1 - y_i)$ for which eq. (1.1) holds. Indeed, this logarithmic preprocessing step is built into commercial CT scanners (Fu et al., 2016), though some additional preprocessing for calibration and denoising is often performed before the logarithm. The logarithm is well-conditioned for $y_i \approx 0$ but becomes numerically unstable as $y_i$ approaches unity, which corresponds to total X-ray absorption. This is particularly problematic for rays that pass through high-density materials, such as metal, as well as for very low-dose CT scans that use fewer X-ray photons.

Instead, we study reconstruction through direct inversion of eq. (1.2) via iterative gradient descent. We optimize a squared loss function over these nonlinear measurements $y_i$, which is optimal for the case of Gaussian measurement noise—though extending this analysis to more realistic noise models is also of interest (Fu et al., 2016). By avoiding the ill-conditioned logarithm of a near-zero measurement, this approach is well-suited to CT reconstruction with low-dose X-rays as well as CT reconstruction with reduced metal artifacts. However, direct reconstruction through eq. (1.2) is more challenging than reconstruction through the linearized eq. (1.1), because the resulting loss function is nonconvex. While the linear inverse problem defined by eq. (1.1) can be solved in closed form by methods such as Filtered Back Projection (Radon, 1917; Kak & Slaney, 2001; Natterer, 2001), the nonlinear inverse problem defined by eq. (1.2) requires an iterative solution to a nonconvex optimization problem. We show that gradient descent with appropriate stepsize successfully recovers the global optimum of this nonconvex objective, suggesting that direct optimization through eq. (1.2) is a viable and desirable alternative to current methods that use eq. (1.1).

Concretely, we make the following contributions:

- We propose a Gaussian model of the $\boldsymbol{a}_i$ in eq. (1.2) and show that gradient descent converges to the global optimum of this model at a geometric rate, despite the nonconvex formulation and with a near minimal number of random measurements. To prove this result we utilize and build upon intricate arguments for uniform concentration of empirical processes.

- We extend our result to a compressive sensing setting to show that a structured signal can be recovered from far fewer measurements than its dimension. In this case prior information of the signal structure is enforced via a convex regularizer; our result holds for *any* convex regularizer. We show that the required number of measurements is commensurate to an appropriate notion of statistical dimension that captures how well the regularizer enforces the structural assumptions about the signal. For example, for an $s$-sparse signal and an $\ell_1$ regularizer our results require on the order of $s \log(n/s)$ measurements, where $n$ is the dimension of the signal. This is the optimal sample complexity even for linear measurements.

- We perform an empirical comparison of reconstruction quality in 3D cone-beam CT on both synthetic and real volumes, where our real dataset consists of a human skull with metal dental crowns. We show that direct reconstruction through eq. (1.2) yields reduction in metal artifacts compared to reconstruction by inverting the nonlinearity into eq. (1.1).

## 2 PROBLEM FORMULATION

In practice, the measurement vectors $\boldsymbol{a}_i$ are sparse, nonnegative, highly structured, and dependent on the rays $i$, as only a small subset of signal values in $\boldsymbol{x}$ will contribute to any particular ray. These vectors correspond to the weights in a discretized ray integral (projection) along the $i$'th measurement ray in the Radon transform (Radon, 1917). We use these real, ray-structured measurement vectors in our synthetic and real-data experiments.

In our theoretical analysis, we make two simplifying alterations to eq. (1.2): (1) we model $\boldsymbol{a}_i$ as a standard Gaussian vector, where the Gaussian randomness is an approximation of the randomness in the choice of ray direction, and (2) we wrap the inner product $\boldsymbol{a}_i^T \boldsymbol{x}$ in a ReLU, to capture the physical reality that the raw integral of density along a ray, and the corresponding sensor measurement, must always be nonnegative. This nonnegativity is implicit in eq. (1.2) because $\boldsymbol{a}_i$ represents a ray integral with only nonnegative weights as its entries, and the true density signal $\boldsymbol{x}$ is also nonnegative; in our model eq. (2.1) we make nonnegativity explicit (subscript $+$ denotes ReLU):

$$y_i = f(\boldsymbol{a}_i^T \boldsymbol{x}), \text{ where } f(\cdot) = 1 - \exp(-(\cdot)_+). \tag{2.1}$$

Here $y_i \in \mathbb{R}$ is a measurement corresponding to ray $i$, $\boldsymbol{x} \in \mathbb{R}^n$ is the signal we want to recover, and $\boldsymbol{a}_i \in \mathbb{R}^n$ are i.i.d. random Gaussian measurement vectors distributed as $\mathcal{N}(\boldsymbol{0}, \boldsymbol{I}_n)$. In this paper we consider a least-squares loss of the form

$$\mathcal{L}(\boldsymbol{z}) = \frac{1}{2m} \sum_{i=1}^{m} (y_i - f(\boldsymbol{a}_i^T \boldsymbol{z}))^2, \tag{2.2}$$

which is optimal in the presence of Gaussian measurement noise. However, in our analysis we focus on the noiseless setting. We minimize this loss using subgradient descent starting from $\boldsymbol{z}_0 = \boldsymbol{0}_n$, with step size $\mu_t$ in step $t$. More specifically, the iterates take the form

$$\boldsymbol{z}_t = \boldsymbol{z}_{t-1} - \mu_t \nabla \mathcal{L}(\boldsymbol{z}_{t-1}) = \boldsymbol{z}_{t-1} - \frac{\mu_t}{m} \sum_{i=1}^{m} \boldsymbol{a}_i f'(\boldsymbol{a}_i^T \boldsymbol{z}_{t-1})(f(\boldsymbol{a}_i^T \boldsymbol{z}_{t-1}) - y_i).$$

Here, we use the following subdifferential of $f$:

$$f'(\cdot) = \begin{cases} 0, & \text{if } \cdot < 0 \\ \frac{1}{2}, & \text{if } \cdot = 0 \\ \exp(-\cdot), & \text{if } \cdot > 0 \end{cases}.$$

We also consider a regularized (compressive sensing) setting where the number of measurements $m$ is significantly smaller than the dimension $n$ of the signal. In this case we optimize the augmented loss function

$$\mathcal{L}(\boldsymbol{z}) = \frac{1}{2m} \sum_{i=1}^{m} (y_i - f(\boldsymbol{a}_i^T \boldsymbol{z}))^2 + \lambda \mathcal{R}(\boldsymbol{z}) \tag{2.3}$$

via subgradient descent. Here, $\mathcal{R}(\boldsymbol{z})$ is a regularizer enforcing *a priori* structure about the signal, with regularization weight $\lambda$. In our experiments, we use 3D total variation as $\mathcal{R}$, to encourage our reconstructed structure to have sparse gradients in 3D space.

We note that $\mathcal{L}$ is a nonconvex objective, so it is not obvious whether or not subgradient descent will reach the global optimum. Do the iterates converge to the correct solution? How many iterations are required? How many measurements? How does the number of measurements depend on the signal structure and the choice of regularizer? In the following sections, we take steps to answer these questions.

**Connecting Theory to Practice.** Our theoretical modeling assumption of $\boldsymbol{a}_i$ as standard Gaussian rather than a highly structured projection is based upon two approximation steps, each of which is well-supported in the literature. A projection in real-space is equivalent to a slice through the origin in Fourier space, according to the Fourier slice theorem. Our first approximation involves sampling uniformly random Fourier coefficients rather than radial slices in Fourier space. This approximation is common in the literature, for example in the seminal compressive sensing paper by Candès, Romberg, and Tao Candes et al. (2006). Our second approximation is to replace random Fourier measurements with random Gaussian measurements. This approximation is also well supported by prior work, namely Oymak et al. (2018), which shows that "structured random matrices behave similarly to random Gaussian matrices" in compressive sensing type settings.

We have also verified that the key pseudoconvexity property we use in our proofs—namely, the strict positivity of the correlation quantity $\nabla\mathcal{L}(z)^T(z-x)/\|z-x\|_{\ell_2}^2$, where $x$ is the true signal and $z$ is the current iterate—holds in practice even when we evaluate it using the true projection-based loss function instead of our Gaussian model. Figure 1 shows a plot of this correlation quantity at various distances between $z$ and $x$, where the x axis is normalized by $\|x\|_{\ell_2} \approx 7.4$ so that the origin (which is also the initialization value of $z$) has normalized distance 1 relative to the global optimum $x$. We sample 100 values of $z$ at various distances from $x$, and show that the correlation is empirically positive across all of the distances gradient descent would encounter from initialization to convergence. This experiment uses the true forward model that respects the ray structure, verifying that the same pseudoconvexity property we show in our Gaussian model also holds in the full forward model.

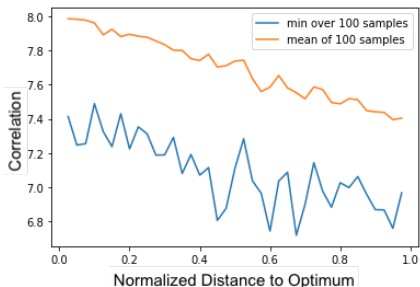

Figure 1: The pseudoconvexity property central to our proofs in the Gaussian model also holds empirically in the full ray-structured model, shown here via positivity of the correlation quantity $\nabla\mathcal{L}(z)^T(z-x)/\|z-x\|_{\ell_2}^2$.

## 3 GLOBAL CONVERGENCE IN THE UNREGULARIZED SETTING

Our first result shows that in the unregularized setting, direct gradient-based updates converge globally at a geometric rate. We defer the proof of Theorem 1 to Appendix B.

**Theorem 1** *Consider the problem of reconstructing a signal $x \in \mathbb{R}^n$ from $m$ nonlinear CT measurements of the form $y_i = 1 - e^{-(a_i^T x)_+}$, where the measurement vectors $a_i$ are generated i.i.d. $\mathcal{N}(\mathbf{0}, I_n)$. We consider a least-squares loss as in eq. (2.2) and run gradient updates of the form*

$$z_t = z_{t-1} - \mu_t \nabla\mathcal{L}(z_{t-1})$$

*starting from $z_0 = \mathbf{0}_n$ with $\mu_1 = 4\exp\left(-\frac{\|x\|_{\ell_2}^2}{2}\right)\frac{1}{\mathrm{erfc}\left(\frac{\|x\|_{\ell_2}}{\sqrt{2}}\right)}$ and $\mu_t = \mu e^{-5\|x\|_{\ell_2}}$ with $\mu \leq c_0$*

*for $t > 1$. Here, $\mathrm{erfc}$ is the complementary error function. As long as the number of measurements obeys*

$$m \geq \frac{c_1 e^{c_2\|x\|_{\ell_2}}}{\|x\|_{\ell_2}^2}n$$

*then*

$$\|z_t - x\|_{\ell_2}^2 \leq \left(1 - \mu e^{-10\|x\|_{\ell_2}}\right)^t \|x\|_{\ell_2}^2$$

*holds with probability at least $1 - 5e^{-c_3 n} - 3e^{-\frac{m}{2}}$. Here, $c_0, c_1, c_2$, and $c_3$ are fixed positive numerical constants.*

Theorem 1 answers some of the key questions from the previous section in the affirmative. Even though the nonlinear CT reconstruction problem is a nonconvex optimization, gradient descent converges to the global optimum, the true signal, at a geometric rate.

Further, the number of required measurements $m$ is on the order of $n$, the dimension of the signal, which is near-minimal even for a linear forward model. In Theorem 2 we prove global convergence with even fewer measurements in the compressive sensing setting, when some prior knowledge of the signal structure is enforced through a convex regularizer.

We note that the initial step size $\mu_1$ used in Theorem 1 is a function of the signal norm $\|x\|_{\ell_2}$, which is *a priori* unknown. However, we briefly describe how this quantity can be estimated from the available measurements. By averaging over the $m$ measurements, we have

$$\frac{1}{m}\sum_{i=1}^{m} y_i = \frac{1}{m}\sum_{i=1}^{m}(1 - e^{-(a_i^T x)_+}) = 1 - \frac{1}{m}\sum_{i=1}^{m} e^{-g_i + \|x\|_{\ell_2}}$$

where $g_i$ are i.i.d. standard Gaussian random variables. Since $e^{-g_i + \|x\|_{\ell_2}}$ is a 1-Lipschitz function of $g_i$, this quantity concentrates around its mean

$$\mathbb{E}\left[ e^{-g_i + \|x\|_{\ell_2}} \right] = \frac{1}{2} \left( 1 + \exp\left( \frac{\|x\|_{\ell_2}^2}{2} \right) \operatorname{erfc}\left( \frac{\|x\|_{\ell_2}}{\sqrt{2}} \right) \right).$$

We can invert this relationship to get a close estimate of $\|x\|_{\ell_2}$ from the average measurement value.

We also note that both the convergence rate and the number of measurements in Theorem 1 are exponentially dependent on $\|x\|_{\ell_2}$. This is natural because as $\|x\|_{\ell_2}$ increases towards infinity the measurements $y_i = 1 - e^{-(a_i^T x)_+}$ approach the constant value 1 and the corresponding gradient of the loss approaches zero. Intuitively, this corresponds to trying to recover a CT scan of a metal box; if the walls of the box become infinitely absorbing of X-rays, we cannot hope to see inside it. Nonetheless, for real and realistic metal components in our experiments (Section 5) we do find good signal recovery following this approach.

## 4 GLOBAL CONVERGENCE IN THE REGULARIZED SETTING

We now turn our attention to the regularized setting. Our measurements again take the form $y_i = 1 - e^{-(a_i^T x)_+}$ for $i = 1, 2, \ldots, m$, where $x \in \mathbb{R}^n$ is the unknown but now *a priori* "structured" signal. In this case we wish to use many fewer measurements $m$ than the number of variables $n$, to reduce the X-ray exposure to the patient without sacrificing the resolution of the reconstructed image or volume $x$. Because the number of equations $m$ is significantly smaller than the number of variables $n$, there are infinitely many reconstructions obeying the measurement constraints. However, it may still be possible to recover the original signal by exploiting knowledge of its structure. To this aim, let $\mathcal{R} : \mathbb{R}^n \to \mathbb{R}$ be a regularization function that reflects some notion of "complexity" of the "structured" solution. For the sake of our theoretical analysis we will use the following constrained optimization problem in lieu of eq. (2.3) to recover the signal:

$$\min_{z \in \mathbb{R}^n} \mathcal{L}(z) = \frac{1}{2m} \sum_{i=1}^m (y_i - f(a_i^T z))^2 \quad \text{subject to} \quad \mathcal{R}(z) \leq \mathcal{R}(x). \tag{4.1}$$

We solve this optimization problem using projected gradient updates of the form

$$z_{t+1} = \mathcal{P}_{\mathcal{K}}\left( z_t - \mu_{t+1} \nabla \mathcal{L}(z_t) \right). \tag{4.2}$$

Here, $\mathcal{P}_{\mathcal{K}}(z)$ denotes the projection of $z \in \mathbb{R}^n$ onto the constraint set

$$\mathcal{K} = \{ z \in \mathbb{R}^n : \mathcal{R}(z) \leq \mathcal{R}(x) \}. \tag{4.3}$$

We wish to characterize the rate of convergence of the projected gradient updates eq. (4.2) as a function of the number of measurements, the available prior knowledge of the signal structure, and how well the choice of regularizer encodes this prior knowledge. For example, if our unknown signal $x$ is approximately sparse, using an $\ell_1$ norm for the regularizer is superior to using an $\ell_2$ regularizer. To make these connections precise and quantitative, we need a few definitions which we adapt verbatim from Oymak et al. (2017); Oymak & Soltanolkotabi (2017b); Soltanolkotabi (2019b).

**Definition 1 (Descent set and cone)** *The* set of descent *of a function $\mathcal{R}$ at a point $x$ is defined as*

$$\mathcal{D}_{\mathcal{R}}(x) := \left\{ h : \ \mathcal{R}(x + h) \leq \mathcal{R}(x) \right\}.$$

*The* cone of descent, *or* tangent cone, *is the conic hull of the descent set, or the smallest closed cone $\mathcal{C}_{\mathcal{R}}(x)$ that contains the descent set, i.e. $\mathcal{D}_{\mathcal{R}}(x) \subset \mathcal{C}_{\mathcal{R}}(x)$.*

The size of the descent cone $\mathcal{C}_{\mathcal{R}}$ determines how well the regularizer $\mathcal{R}$ captures the structure of the unknown signal $x$. The smaller the descent cone, the more precisely the regularizer describes the properties of the signal. We quantify the size of the descent cone using mean (Gaussian) width.

**Definition 2 (Gaussian width)** *The Gaussian width of a set $\mathcal{C} \in \mathbb{R}^p$ is defined as:*

$$\omega(\mathcal{C}) := \mathbb{E}_g[\sup_{z \in \mathcal{C}} \langle g, z \rangle],$$

*where the expectation is taken over $g \sim \mathcal{N}(0, I_p)$.*

We now have all the definitions in place to quantify how well $\mathcal{R}$ captures the properties of the unknown signal $\boldsymbol{x}$. This leads us to the definition of the minimum required number of measurements.

**Definition 3 (minimal number of measurements)** *Let $\mathcal{C}_{\mathcal{R}}(\boldsymbol{z})$ be a cone of descent of $\mathcal{R}$ at $\boldsymbol{z}$. We define the minimal sample function as*

$$\mathcal{M}(\mathcal{R}, \boldsymbol{z}) := \omega^2(\mathcal{C}_{\mathcal{R}}(\boldsymbol{z}) \cap \mathcal{B}^n),$$

*where we use $\mathcal{B}^n$ to denote the the unit ball of $\mathbb{R}^n$. We shall often use the short hand $m_0 = \mathcal{M}(\mathcal{R}, \boldsymbol{z})$ with the dependence on $\mathcal{R}, \boldsymbol{z}$ implied. Here we define $m_0$ for an arbitrary point $\boldsymbol{z}$, but we will apply the definition at the signal $\boldsymbol{x}$.*

We note that $m_0$ is exactly the minimum number of samples required for structured signal recovery from *linear* measurements when using convex regularizers. Specifically, the optimization problem

$$\arg \min_{\boldsymbol{z}} \frac{1}{2m} \sum_{i=1}^m \left( y_i - \boldsymbol{a}_i^T \boldsymbol{z} \right)^2 \quad \text{subject to} \quad \mathcal{R}(\boldsymbol{z}) \leq \mathcal{R}(\boldsymbol{x}), \tag{4.4}$$

succeeds at recovering the unknown signal $\boldsymbol{x}$ with high probability from $m$ measurements of the form $y_i = \boldsymbol{a}_i^T \boldsymbol{x}$ if and only if $m \geq m_0$. We note that $m_0$ only approximately characterizes the minimum number of samples required. A more precise characterization is $\phi^{-1}(\omega^2(\mathcal{C}_{\mathcal{R}}(\boldsymbol{x}) \cap \mathcal{B}^n)) \approx \omega^2(\mathcal{C}_{\mathcal{R}}(\boldsymbol{x}) \cap \mathcal{B}^n)$ where $\phi(t) = \sqrt{2} \frac{\Gamma\left(\frac{t+1}{2}\right)}{\Gamma\left(\frac{t}{2}\right)} \approx \sqrt{t}$, where we use $\mathcal{B}^n$ to denote the the unit ball of $\mathbb{R}^n$. However, since our results have unspecified constants we avoid this more accurate characterization. Given that in our Gaussian-approximated nonlinear CT reconstruction problem we have less information (we lose information when the input to the ReLU is negative), we cannot hope to recover structured signals from $m \leq m_0$ when using (4.1). Therefore, we can use $m_0$ as a lower-bound on the minimum number of measurements required for projected gradient descent iterations eq. (4.2) to succeed in recovering the signal of interest. With these definitions in place we are now ready to state our theorem in the regularized/compressive sensing setting. We defer the proof of Theorem 2 to Appendix C.

**Theorem 2** *Consider the problem of reconstructing a signal $\boldsymbol{x} \in \mathbb{R}^n$ from $m$ nonlinear CT measurements of the form $y_i = 1 - e^{-(\boldsymbol{a}_i^T \boldsymbol{x})_+}$, where the measurement vectors $\boldsymbol{a}_i$ are generated i.i.d. $\mathcal{N}(\boldsymbol{0}, \boldsymbol{I}_n)$. We consider a constrained least-squares loss as in eq. (4.1) and run projected gradient updates of the form in eq. (4.2) starting from $\boldsymbol{z}_0 = \boldsymbol{0}_n$ with $\mu_1 = 4 \exp\left(-\frac{\|\boldsymbol{x}\|_{\ell_2}^2}{2}\right) \frac{1}{\operatorname{erfc}\left(\frac{\|\boldsymbol{x}\|_{\ell_2}}{\sqrt{2}}\right)}$ and $\mu_t = \mu e^{-5\|\boldsymbol{x}\|_{\ell_2}}$ with $\mu \leq \frac{c_0}{\left(1 + \frac{n}{m}\right)^2}$ for $t > 1$. Here, $\operatorname{erfc}$ is the complementary error function. As long as the number of measurements obeys*

$$m \geq \frac{c_1 e^{c_2 \|\boldsymbol{x}\|_{\ell_2}}}{\|\boldsymbol{x}\|_{\ell_2}^2} m_0,$$

*with $m_0$ denoting the minimal number of samples per Definition 3, then*

$$\|\boldsymbol{z}_t - \boldsymbol{x}\|_{\ell_2}^2 \leq \left(1 - \mu e^{-10\|\boldsymbol{x}\|_{\ell_2}}\right)^t \|\boldsymbol{x}\|_{\ell_2}^2$$

*holds with probability at least $1 - 5e^{-c_3 m_0} - 3e^{-\frac{m}{2}}$. Here, $c_0, c_1, c_2,$ and $c_3$ are fixed positive numerical constants.*

Theorem 2 parallels Theorem 1, showing fast geometric convergence to the global optimum despite nonconvexity. In this regularized setting, the sample complexity of our nonlinear reconstruction problem is on the order of $m_0$, the number of measurements required for linear compressive sensing. In other words, the number of measurements required for regularized nonlinear CT reconstruction from raw measurements is within a constant factor of the number of measurements needed for the same reconstruction from linearized measurements. This is the optimal sample complexity for this nonlinear reconstruction task. For instance for an $s$ sparse signal for which $m_0 \propto s \log(n/s)$, Theorem 2 states that on the order of $s \log(n/s)$ nonlinear CT measurements suffices for our direct gradient-based approach to succeed. Finally, we would like emphasize that the above result is rather general as it applies to any type structure in the signal and can also deal with any convex regularizer.

## 5 EXPERIMENTS

We support our theoretical analysis with experimental evidence that gradient-based optimization through the nonlinear CT forward model is effective for a wide range of signal densities, including signals that are dense enough that the same optimization procedure through the linearized forward model produces noticeable "metal artifacts." All of our experiments are based on the JAX implementation of Plenoxels (Sara Fridovich-Keil and Alex Yu et al., 2022), with a dense 3D grid of optimizable density values connected by trilinear interpolation. We use a cone-beam CT (CBCT) setup and optimize with mild total variation regularization. Our experiments do not focus on speed or measurement sparsity, though we expect our optimization to pair naturally with efficient ray sampling implementations and regularizers of choice.

**Synthetic Data.** Our synthetic experiments use a ground truth volume defined by the standard Shepp-Logan phantom (Shepp & Logan, 1974) in 3D, with the following modifications: (1) we scale down the voxel density values by a factor of 4, to more closely mimic the values in our real CBCT skull dataset, and (2) we adjust one of the ellipsoids to be slightly larger than standard (to make it more visible), and gradually increase its ground truth density to simulate a spectrum from soft tissue to bone to metal. We simulate CT observations of this synthetic volume and then reconstruct using either the linearized forward model with the logarithm and eq. (1.1), or directly using eq. (1.2). We also use a small amount of total variation regularization for both linearized and nonlinear reconstruction, and constrain results to be nonnegative.

Results of this synthetic experiment are presented in Figure 2. As the density of the test ellipsoid increases, the linearized reconstruction experiences increasingly severe "metal artifacts," while the nonlinear reconstruction continues to closely match the ground truth. PSNR values are reported over the entire reconstructed volume compared to the ground truth, where PSNR is defined as $-10 \log_{10}(\mathrm{MSE})$ and MSE is the mean squared voxel-wise error. Note that this synthetic experiment does not include any measurement noise or miscalibration; the instability of the logarithm with respect to dense signals arises even when the only noise is due to numerical precision. We also note that even the densest synthetic "metal" ellipsoid we test is no denser than what we observe in the metal dental crown in our real CBCT skull dataset in Figure 3.

**Real Data.** Our real data experiment uses CBCT measurements of a human skull with metal dental crowns on some of the teeth. In Figure 3 we show slices of our nonlinear reconstruction compared to a reference state-of-the-art commercial linearized reconstruction, as no ground truth is available for the real volume. We also compare with a standard linearized baseline, FDK Biguri et al. (2016), which exhibits severe artifacts especially around the metal crown.

## 6 RELATED WORK

**Tomographic reconstruction.** The measurement model in eq. (1.2) is a discretized corollary of the Beer-Lambert Law that governs the attenuation of light as it passes through absorptive media. Inverting the exponential nonlinearity in this model recovers the Radon transform summarized by eq. (1.1), in which measurements are linear projections of the signal at chosen measurement angles. The Radon transform has a closed-form inverse transform, Filtered Back Projection (FBP) (Radon, 1917; Kak & Slaney, 2001; Natterer, 2001), that leverages the Fourier slice theorem (Bracewell, 1990; Kak & Slaney, 2001). FBP is well-understood and can be computed efficiently, and is a standard option in commercial CT scanners, but its reconstruction quality can suffer in the presence of either limited measurement angles or metal (highly absorptive) signal components (Fu et al., 2016).

Many methods exist to improve the quality of CT reconstruction in the limited-measurement regime, which is of clinical interest because every additional measurement exposes the patient to ionizing X-ray radiation. These methods typically involve augmenting the data-fidelity loss function with a regularization term that describes some prior knowledge of the signal to be reconstructed. Such priors include sparsity (implemented through an $\ell_1$ norm) in a chosen basis, such as wavelets (Foucart & Rauhuti, 2013; Chambolle et al., 1998), as well as gradient sparsity (implemented through total variation regularization) (Candes et al., 2006). Compressive sensing theory guarantees correct recovery with fewer measurements in these settings, as long as the true signal is well-described by the chosen prior (Foucart & Rauhuti, 2013). CT reconstruction with priors cannot be solved in closed form, but as

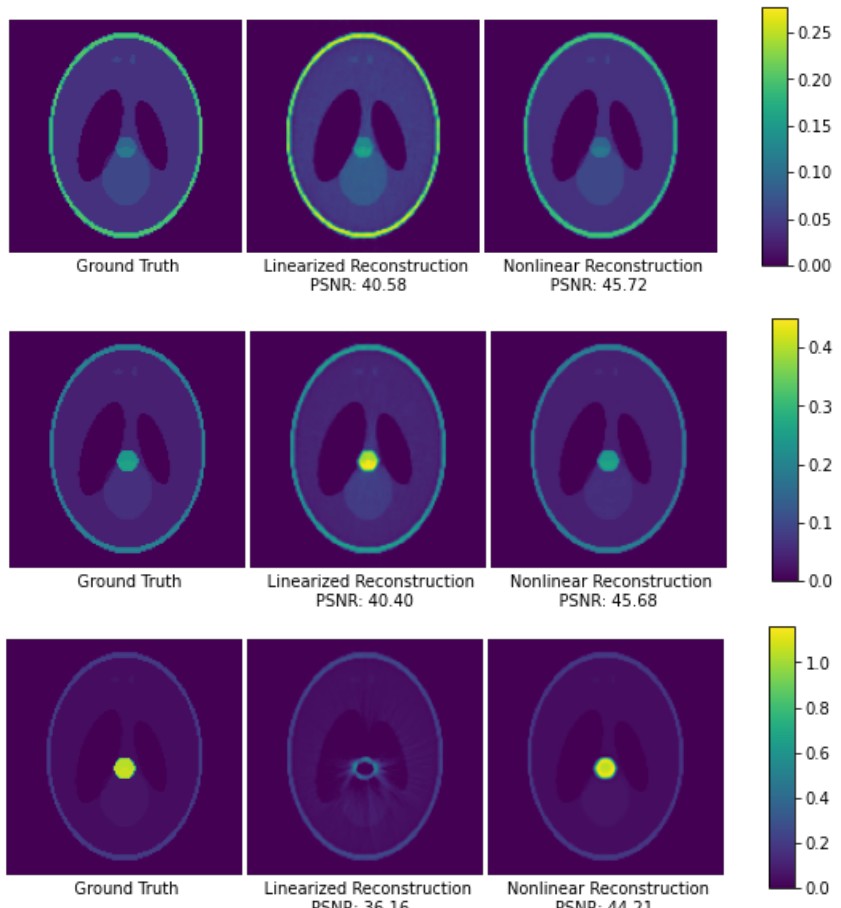

Figure 2: Synthetic experiments using the Shepp-Logan phantom, showing a slice through the reconstructed 3D volume. From top to bottom, we increase the density of the central test ellipsoid to simulate soft tissue, bone, and metal. Nonlinear reconstruction is robust even to dense "metal" elements of the target signal.

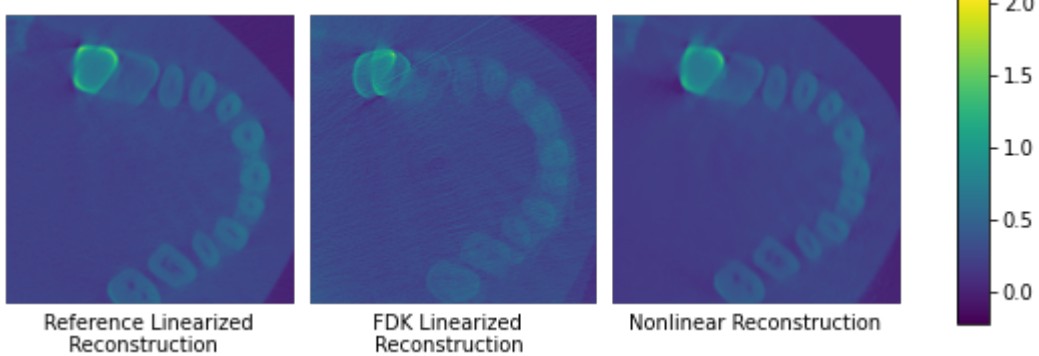

Figure 3: Real experiments using a human skull with a metal dental crown, showing a slice of the reconstructed volume. Note the streak artifact to the left of the metal crown in the reference linearized reconstruction, and the streak below the crown (and misshapen crown) in the FDK reconstruction, which are not present in the nonlinear reconstruction.

long as the regularization is convex we are guaranteed that iterative optimization methods such as gradient descent, ISTA (Chambolle et al., 1998), and FISTA (Beck & Teboulle, 2009) will be successful.

Recently, reconstruction with even fewer measurements has been proposed by leveraging deep learning, through either neural scene representation (Rückert et al., 2022) or data-driven priors (Szczykutowicz et al., 2022). These methods may sacrifice convexity, and theoretical guarantees, in favor of more flexible and adaptive regularization that empirically reduces reconstruction artifacts in the limited-measurement regime. However, these methods are still based on the linear measurement model of eq. (1.1), making them susceptible to reconstruction artifacts near highly absorptive metal components. In some cases neural methods may reduce metal artifacts compared to traditional algorithms, but this reduction is achieved by leveraging strong and adaptive prior knowledge, meaning any improvements may "stack" with use of the nonlinear measurement model.

Our method may pair particularly well with new photon-counting CT scanners (Shikhaliev et al., 2005), which were approved by the FDA in 2021 (Food & Administration, 2021). These scanners measure raw X-ray photon counts, which should enable finer-grained noise modeling and correction as well as our nonlinear method for principled reconstruction of signals with metal.

**Signal reconstruction from nonlinear measurements.** There are a growing number of papers focused on reconstructing a signal from nonlinear measurements or single index models. Early papers on this topic focus on phase retrieval and ReLU nonlinearities (Oymak et al., 2018; Soltanolkotabi, 2017; Candes et al., 2015) and approximate reconstruction (Oymak & Soltanolkotabi, 2017a), but do not handle the compressive sensing/structured signal reconstruction setting. Soltanolkotabi (2019a) deals with reconstruction from structured signals for intensity and absolute value nonlinearities but only achieves the optimal sample complexity locally. A more recent paper (Mei et al., 2018) deals with a variety of nonlinearities with bounded derivative activations, but does not handle non-differentiable activations and only deals with simple structured signals such as sparse ones. In contrast, our activation is non-differentiable, we handle arbitrary structures in the signal, and our results apply for any convex regularizer.

## 7 DISCUSSION

In this paper, we consider the CT reconstruction problem from raw nonlinear measurements of the form $y_i = 1 - e^{-\boldsymbol{a}_i^T \boldsymbol{x}}$ for a signal $\boldsymbol{x}$ and random measurement weights $\boldsymbol{a}_i$. Although this nonlinear measurement model can be easily transformed into a linear model via a logarithmic preprocessing step $\hat{y}_i = -\ln(1 - y_i) = \boldsymbol{a}_i^T \boldsymbol{x}$, and this transformation is common practice in clinical CT reconstruction, the logarithm is numerically unstable when the measurements approach unity. This occurs frequently in practice, notably when the signal $\boldsymbol{x}$ contains metal and especially for low-dose CT scanners that reduce radiation exposure. In this setting, traditional linear reconstruction methods tend to produce "metal artifacts" such as streaks around metal implants. Reconstruction directly through the raw nonlinear measurements avoids this numerically unstable preprocessing, in exchange for solving a nonconvex nonlinear least squares objective instead of convex linear least squares.

We prove that gradient descent finds the global optimum in CT reconstruction from raw nonlinear measurements, recovering exactly the true signal $\boldsymbol{x}$ despite the nonconvex optimization. Moreover, it converges at a geometric rate, which is considered fast even for convex optimization. This nonconvex optimization requires order $n$ measurements, where $n$ is the dimension of the unknown signal, the same order sample complexity as if we had reconstructed through a linear forward model. We also extend our theoretical results to the compressive sensing setting, in which prior structural knowledge of the signal $\boldsymbol{x}$, enforced through a regularizer, allows for reconstruction with far fewer measurements than $n$. Our results in this setting again parallel standard results from the linear reconstruction problem, even though we consider a nonlinear forward model and optimize a nonconvex formulation.

We also compare linearized and nonlinear CT reconstruction experimentally in the setting of 3D cone-beam CT, using both a synthetic 3D Shepp-Logan phantom for which we know the ground truth volume as well as a real human skull with metal dental crowns. In both cases, we find that nonlinear reconstruction reduces metal artifacts compared to linearized reconstruction, whether that linearized reconstruction is done by gradient descent or a commercial algorithm. Our work is a promising first step towards higher-quality CT reconstruction in the presence of metal components and low-dose X-rays, offering both practical and theoretical guidance for trustworthy reconstruction. Future work may extend our results both theoretically and experimentally to consider more realistic measurement noise settings such as Poisson noise, which is particularly timely given the emergence of new photon-counting CT scanners.

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

## A  APPENDIX

## B  PROOF OF THEOREM 1

In the following subsections we prove Theorem 1, beginning with a proof outline.

### B.1  PROOF OUTLINE

The proof consists of the following four steps.

**Step I: First iteration: Welcome to the neighborhood.**
In the first step, we show that as long as the number of measurements is sufficiently large ($m \geq 5n$), the first iteration obeys

$$\|\boldsymbol{z}_1 - \boldsymbol{x}\|_{\ell_2} \leq \frac{1}{4}\|\boldsymbol{x}\|_{\ell_2} \tag{B.1}$$

with probability at least $1 - 2e^{-\frac{m}{2}} - e^{-cn}$, for a constant $c$. We prove this in Appendix B.2.

**Step II: Local pseudoconvexity.**

In the second step, we show that our nonconvex objective function is locally strongly pseudoconvex inside this local neighborhood of eq. (B.1). Specifically, we show the correlation inequality

$$\nabla \mathcal{L}(\boldsymbol{z})^T(\boldsymbol{z} - \boldsymbol{x}) \geq \alpha \|\boldsymbol{z} - \boldsymbol{x}\|_{\ell_2}^2 \tag{B.2}$$

holds with $\alpha = \frac{1}{2}e^{-(5\|\boldsymbol{x}\|_{\ell_2}+2)}$, with probability at least $1 - 4e^{-n}$ as long as the number of measurements $m$ is at least $\frac{c_1}{e^{c_2\|\boldsymbol{x}\|_{\ell_2}}\|\boldsymbol{x}\|_{\ell_2}^2}n$ for constants $c_1$ and $c_2$.

We compute the $\alpha$ in Equation (B.2) in two cases, where case 1 corresponds to $\frac{\boldsymbol{x}^T(\boldsymbol{z}-\boldsymbol{x})}{\|\boldsymbol{x}\|_{\ell_2}\|\boldsymbol{z}-\boldsymbol{x}\|_{\ell_2}} \geq -0.6$ and case 2 corresponds to $\frac{\boldsymbol{x}^T(\boldsymbol{z}-\boldsymbol{x})}{\|\boldsymbol{x}\|_{\ell_2}\|\boldsymbol{z}-\boldsymbol{x}\|_{\ell_2}} < -0.6$. These cases are analyzed in Appendix B.3 and Appendix B.4, respectively. Combining cases 1 and 2, we have that the lower bound on $\alpha$ from case 2 lower bounds the bound from case 1, as shown in Figure 4, so we use that bound in eq. (B.2). The

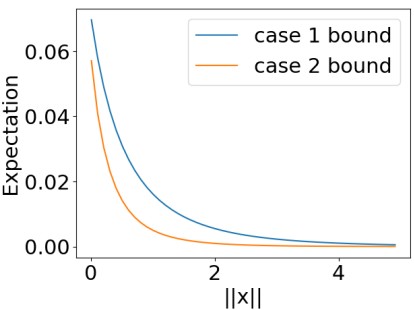

Figure 4: Case 2 provides a lower bound on the expected correlation for both cases.

sample complexity in eq. (B.2) is the maximum over the sample complexities of cases 1 and 2, which are $m \geq \frac{cn}{\alpha^2\|\boldsymbol{x}\|_{\ell_2}^2}$ and $m \geq \frac{Cn}{(\sqrt{2\alpha}-\sqrt{\alpha})^2}$, respectively, for constants $c$ and $C$.

**Step III: Smoothness.**
In the third step, we show

$$\|\nabla \mathcal{L}(\boldsymbol{z})\|_{\ell_2} \leq L\|\boldsymbol{z} - \boldsymbol{x}\|_{\ell_2} \tag{B.3}$$

holds with probability at least $1 - e^{-\frac{1}{2}(m+n)}$, for a constant $L$. This smoothness condition is proved in Appendix B.5.

**Step IV: Completing the proof via combining Steps I-III.**
Finally, we combine the first three steps into a complete proof. At the core of the proof is the following lower bound on the correlation between the loss gradient and the error vector

$$\nabla \mathcal{L}(\boldsymbol{z})^T(\boldsymbol{z} - \boldsymbol{x}) \geq A\|\boldsymbol{z} - \boldsymbol{x}\|_{\ell_2}^2 + B\|\nabla \mathcal{L}(\boldsymbol{z})\|_{\ell_2}^2 \tag{B.4}$$

for positive constants $A$ and $B$. This starting point is similar to the proof of Lemma 7.10 in Candes et al. (2015), with some modifications necessary for the nonlinear CT reconstruction problem. To prove eq. (B.4) we first combine eq. (B.2) and eq. (B.3) to conclude that

$$\nabla \mathcal{L}(\boldsymbol{z})^T(\boldsymbol{z} - \boldsymbol{x}) \geq \frac{\alpha}{2}\|\boldsymbol{z} - \boldsymbol{x}\|_{\ell_2}^2 + \frac{\alpha}{2}\|\boldsymbol{z} - \boldsymbol{x}\|_{\ell_2}^2$$

$$\geq \frac{\alpha}{2}\|\boldsymbol{z} - \boldsymbol{x}\|_{\ell_2}^2 + \frac{\alpha}{2L^2}\|\nabla \mathcal{L}(\boldsymbol{z})\|_{\ell_2}^2.$$

Thus eq. (B.4) holds with $A = \frac{\alpha}{2}$ and $B = C$ for a new constant $C$, with probability at least $1 - 4e^{-n} - e^{-\frac{1}{2}(m+n)} - 2e^{-\frac{m}{2}} - e^{-cn}$, for a constant $c$ (by a union bound over the first three steps of the proof). Using eq. (B.4) with adequate choice of stepsize suffices to prove geometric convergence.

$$
\begin{aligned}
\|z_{t+1} - x\|_{\ell_2}^2 &= \|z_t - \mu_{t+1}\nabla\mathcal{L}(z_t) - x\|_{\ell_2}^2 \\
&\overset{(a)}{=} \|z_t - x\|_{\ell_2}^2 - 2\mu_{t+1}\nabla\mathcal{L}(z_t)^T(z_t - x) + \mu_{t+1}^2 \|\nabla\mathcal{L}(z_t)\|_{\ell_2}^2 \\
&\overset{(b)}{\leq} (1 - 2\mu_{t+1}A) \|z_t - x\|_{\ell_2}^2 + \mu_{t+1}(\mu_{t+1} - 2B) \|\nabla\mathcal{L}(z_t)\|_{\ell_2}^2 \\
&\overset{(c)}{\leq} (1 - 2\mu_{t+1}A) \|z_t - x\|_{\ell_2}^2 .
\end{aligned}
$$

In (a) we expand the square. In (b) we apply eq. (B.4). In (c) we choose $\mu_{t+1} \in (0, 2B]$, making the second term negative. Applying this relation inductively over $T$ steps of gradient descent yields geometric convergence with rate $1 - 2\mu A$, provided that the step size $\mu_t$ is less than $2B$ for $t > 1$.

The number of measurements $m$ required in theorem 1 is the maximum over the number of samples required for each of the first two steps of the proof. For the first step, $m \geq 5n$ measurements are sufficient to reach our neighborhood of radius $\frac{1}{4} \|x\|_{\ell_2}$. For case 1 in the second step, $m \geq \frac{c_1 n}{\alpha^2 \|x\|_{\ell_2}^2}$ measurements are sufficient for correlation concentration. For case 2 in the second step, $m \geq \frac{c_2 n}{(\sqrt{2\alpha} - \sqrt{\alpha})^2}$ measurements are sufficient for concentration. Maximizing over these bounds yields the result in theorem 1 (note that the constants change during the maximization).

## B.2 First Step: Welcome to the Neighborhood

We first consider what happens in expectation when we take our first gradient step starting from an initialization at $z_0 = 0$. The expectation is over the randomness in the Gaussian measurement vector $a$. We have

$$
\begin{aligned}
\mathbb{E}_a[z_1] &= \mathbf{0}_n - \mu_1 \mathbb{E}_a[\nabla\mathcal{L}(z; a)|_{z=0}] \\
&= -\mu_1 \mathbb{E}_a[a f'(0)(f(0) - y)] \\
&\overset{(a)}{=} -\mu_1 \mathbb{E}_a[-\frac{1}{2}ay] \\
&\overset{(b)}{=} \frac{\mu_1}{2} \mathbb{E}_a[a(1 - \exp(-(a^T x)_+))] \\
&\overset{(c)}{=} -\frac{\mu_1}{2} \mathbb{E}_a[a \exp(-(a^T x)_+))] \\
&\overset{(d)}{=} -\frac{\mu_1}{2} \mathbb{E}_a[\frac{xx^T}{\|x\|_{\ell_2}^2} a \exp(-(a^T x)_+))] \\
&\overset{(e)}{=} -\frac{\mu_1}{2} \frac{x}{\|x\|_{\ell_2}} \mathbb{E}_g[g \exp(-g_+ \|x\|_{\ell_2}))] \\
&\overset{(f)}{=} \frac{\mu_1}{4} \exp\left(\frac{\|x\|_{\ell_2}^2}{2}\right) \text{erfc}\left(\frac{\|x\|_{\ell_2}}{\sqrt{2}}\right) x \\
&\overset{(g)}{=} x.
\end{aligned}
$$

In (a) we evaluate $f(0) = 0$ and $f'(0) = \frac{1}{2}$, where for the latter we use $\frac{1}{2}$ as the sub-differential even though $f$ is nondifferentiable at 0 due to the non-differentiability of ReLU (this choice is also justified as it is the expected gradient of $f$ around a small random initialization around 0). In (b) we plug in the value of the measurement $y$. In (c) we use linearity of expectation and evaluate the first term, $\mathbb{E}_a[a] = \mathbf{0}_n$. In (d) we separate the leading $a$ into components parallel and orthogonal to $x$, and evaluate the expectation of the orthogonal term to zero. In (e) we replace $a^T x$ with $g \|x\|_{\ell_2}$ for a scalar Gaussian $g$, as these have the same distribution. In (f) we evaluate the remaining expectation. In (g) we choose $\mu_1 = 4 \exp\left(-\frac{\|x\|_{\ell_2}^2}{2}\right) \frac{1}{\text{erfc}\left(\frac{\|x\|_{\ell_2}}{\sqrt{2}}\right)}$ so that in expectation, our first step exactly recovers the signal $x$.

Because in practice we do not have access to an expectation over infinite measurements, we also care about the concentration of this first gradient step. This first-step concentration determines the local neighborhood around the signal $\boldsymbol{x}$ that our gradient descent will operate within for the remaining iterations.

$$-\mu_1 \nabla \mathcal{L}(\boldsymbol{z} = \boldsymbol{0}) = \frac{\mu_1}{2m} \sum_{i=1}^{m} \boldsymbol{a}_i (1 - \exp(-(\boldsymbol{a}_i^T \boldsymbol{x})_+))$$

$$\overset{(a)}{=} \frac{\mu_1}{2m} \sum_{i=1}^{m} \frac{\boldsymbol{x} \boldsymbol{x}^T}{\|\boldsymbol{x}\|_{\ell_2}^2} \boldsymbol{a}_i (1 - \exp(-(\boldsymbol{a}_i^T \boldsymbol{x})_+)) + \frac{\mu_1}{2m} \sum_{i=1}^{m} (\mathbb{1} - \frac{\boldsymbol{x} \boldsymbol{x}^T}{\|\boldsymbol{x}\|_{\ell_2}^2}) \boldsymbol{a}_i (1 - \exp(-(\boldsymbol{a}_i^T \boldsymbol{x})_+))$$

$$\overset{(b)}{=} \frac{\mu_1}{2m} \sum_{i=1}^{m} g_i (1 - \exp(-(g_i)_+ \|\boldsymbol{x}\|_{\ell_2})) \frac{\boldsymbol{x}}{\|\boldsymbol{x}\|_{\ell_2}} + \frac{\mu_1}{2m} \sum_{i=1}^{m} (\mathbb{1} - \frac{\boldsymbol{x} \boldsymbol{x}^T}{\|\boldsymbol{x}\|_{\ell_2}^2}) \boldsymbol{a}_i (1 - \exp(-(g_i)_+ \|\boldsymbol{x}\|_{\ell_2}))$$

$$\overset{(c)}{=} \frac{\mu_1}{2m} \sum_{i=1}^{m} g_i (1 - \exp(-(g_i)_+ \|\boldsymbol{x}\|_{\ell_2})) \frac{\boldsymbol{x}}{\|\boldsymbol{x}\|_{\ell_2}} + \frac{\mu_1}{2m} \sqrt{\sum_{i=1}^{m} (1 - \exp(-(g_i)_+ \|\boldsymbol{x}\|_{\ell_2}))^2} \boldsymbol{a}_\perp$$

$$\overset{(d)}{=} \frac{\mu_1}{2m} \sum_{i=1}^{m} \hat{g}_i \frac{\boldsymbol{x}}{\|\boldsymbol{x}\|_{\ell_2}} + \frac{\mu_1}{2m} \sqrt{\sum_{i=1}^{m} \tilde{g}_i} \boldsymbol{a}_\perp.$$

In (a) we separate $\boldsymbol{a}_i$ into components parallel and orthogonal to $\boldsymbol{x}$. In (b) we replace $\boldsymbol{a}_i^T \boldsymbol{x}$ with $\|\boldsymbol{x}\|_{\ell_2} g_i$ for a standard scalar Gaussian $g_i$, as these have the same distribution. In (c) we simplify the second term by rewriting it with a standard Gaussian vector $\boldsymbol{a}_\perp$ with $n-1$ free dimensions (constrained to be orthogonal to $\boldsymbol{x}$), and $\boldsymbol{a}_\perp$ independent of $g_i$ for all $i$. In (d) we write $\hat{g}_i := g_i (1 - \exp(-(g_i)_+ \|\boldsymbol{x}\|_{\ell_2}))$ and $\tilde{g}_i := (1 - \exp(-(g_i)_+ \|\boldsymbol{x}\|_{\ell_2}))^2$.

Note that the first term has expectation $\frac{\mu_1}{4} \exp\left(\frac{\|\boldsymbol{x}\|_{\ell_2}^2}{2}\right) \operatorname{erfc}\left(\frac{\|\boldsymbol{x}\|_{\ell_2}}{\sqrt{2}}\right) \boldsymbol{x} = \boldsymbol{x}$ computed above, and the second term has expectation zero because $\tilde{g}_i$ and $\boldsymbol{a}_\perp$ are independent, and $\boldsymbol{a}_\perp$ is mean zero. The first term is aligned with the signal $\boldsymbol{x}$ but with a scaling factor $\frac{\mu}{2m\|\boldsymbol{x}\|_{\ell_2}} \sum_{i=1}^{m} \hat{g}_i$; we can bound the deviation of this scaling from its mean using Hoeffding's concentration bound for sums of sub-Gaussian random variables. We use the definition of sub-Gaussianity provided by Definition 2.2 in Wainwright (2019), for which $\hat{g}_i$ has sub-Gaussian parameter 1. We omit the leading constant $\frac{\mu}{2m\|\boldsymbol{x}\|_{\ell_2}}$, and apply the Hoeffding bound as presented in Proposition 2.5 in Wainwright (2019) to conclude that

$$\left| \sum_{i=1}^{m} (\hat{g}_i - \mathbb{E}[\hat{g}]) \right| \leq s$$

with probability at least $1 - 2e^{-\frac{s^2}{2m}}$.

The second term is a nonnegative scaling $\frac{\mu}{2m} \sqrt{\sum_{i=1}^{m} \tilde{g}_i}$ times a standard $n$-dimensional Gaussian vector $\boldsymbol{a}_\perp$ with $n-1$ degrees of freedom (because it is orthogonal to $\boldsymbol{x}$). Since $\tilde{g}_i$ is bounded in $[0, 1]$, we can upper bound this scaling by $\frac{\mu}{2\sqrt{m}}$. Then the norm of the random vector $\boldsymbol{a}_\perp$ can also be bounded using Exercise 5.2.4 in Vershynin (2018), to conclude that

$$\|\boldsymbol{a}_\perp\|_{\ell_2} \leq \sqrt{\mathbb{E}[\|\boldsymbol{a}_\perp\|_{\ell_2}^2]} + t \overset{(a)}{=} \sqrt{n-1} + t \overset{(b)}{\leq} (1+t)\sqrt{n}$$

with probability at least $1 - e^{-ct^2 n}$ for a constant $c$, where in (a) we evaluate the expectation of a Gaussian norm with $n-1$ degrees of freedom and in (b) we upper bound $n-1$ by $n$ and do a change of variables to replace $t$ with $t\sqrt{n}$.

Putting these together with a union bound, we have that

$$
\begin{aligned}
\|\boldsymbol{z}_1 - \mathbb{E}[\boldsymbol{z}_1]\|_{\ell_2} = \|\boldsymbol{z}_1 - \boldsymbol{x}\|_{\ell_2} &\leq \frac{\mu_1}{2m}\left|\sum_{i=1}^{m}(\hat{g}_i - \mathbb{E}[\hat{g}])\right| + \frac{\mu_1}{2\sqrt{m}}\|\boldsymbol{a}_\perp\|_{\ell_2} \\
&\leq \frac{\mu_1}{2}\left(\frac{s}{m} + (1+t)\sqrt{\frac{n}{m}}\right) \\
&= 2\exp\left(-\frac{\|\boldsymbol{x}\|_{\ell_2}^2}{2}\right)\frac{1}{\operatorname{erfc}\left(\frac{\|\boldsymbol{x}\|_{\ell_2}}{\sqrt{2}}\right)}\left(\frac{s}{m} + (1+t)\sqrt{\frac{n}{m}}\right) \\
&\stackrel{(a)}{=} 2\exp\left(-\frac{\|\boldsymbol{x}\|_{\ell_2}^2}{2}\right)\frac{1}{\operatorname{erfc}\left(\frac{\|\boldsymbol{x}\|_{\ell_2}}{\sqrt{2}}\right)}\left(s + (1+t)\sqrt{\frac{n}{m}}\right)
\end{aligned}
$$

with probability at least $1 - 2e^{-\frac{s^2 m}{2}} - e^{-ct^2 n}$, where in (a) we change variables and replace $s$ with $sm$, and $c$ is a constant. If we choose $s = t = 1$, this simplifies to

$$
\|\boldsymbol{z}_1 - \boldsymbol{x}\|_{\ell_2} \leq 2\exp\left(-\frac{\|\boldsymbol{x}\|_{\ell_2}^2}{2}\right)\frac{1}{\operatorname{erfc}\left(\frac{\|\boldsymbol{x}\|_{\ell_2}}{\sqrt{2}}\right)}\left(1 + 2\sqrt{\frac{n}{m}}\right)
$$

with probability at least $1 - 2e^{-\frac{m}{2}} - e^{-cn}$, for a constant $c$. For our first step to lie within a distance of $\frac{1}{4}\|\boldsymbol{x}\|_{\ell_2}$, we need the number of measurements to satisfy

$$
m \geq \frac{n}{\left(\frac{\|\boldsymbol{x}\|_{\ell_2}}{16}\exp\left(\frac{\|\boldsymbol{x}\|_{\ell_2}^2}{2}\right)\operatorname{erfc}\left(\frac{\|\boldsymbol{x}\|_{\ell_2}}{\sqrt{2}}\right) - \frac{1}{2}\right)^2}.
$$

The denominator in this expression is lower bounded by $0.2$ for all $\|\boldsymbol{x}\|_{\ell_2}$, so we can also guarantee the first step concentration to this neighborhood using $m \geq 5n$ measurements.

### B.3 CORRELATION CONCENTRATION: CASE 1

Consider the correlation

$$
\nabla\mathcal{L}(\boldsymbol{z})^T(\boldsymbol{z} - \boldsymbol{x}) = \frac{1}{m}\sum_{i=1}^{m}\mathbb{1}_{\{\boldsymbol{a}_i^T\boldsymbol{z}\geq 0\}}e^{-\boldsymbol{a}_i^T\boldsymbol{z}}\left(e^{-(\boldsymbol{a}_i^T\boldsymbol{x})_+} - e^{-\boldsymbol{a}_i^T\boldsymbol{z}}\right)(\boldsymbol{a}_i^T\boldsymbol{h})
$$

where $\boldsymbol{h} := \boldsymbol{z} - \boldsymbol{x}$. Now note that if we have $\boldsymbol{a}_i^T\boldsymbol{x} \geq 0$ and $\boldsymbol{a}_i^T\boldsymbol{h} \geq 0$ it implies $\boldsymbol{a}_i^T\boldsymbol{x} + \boldsymbol{a}_i^T\boldsymbol{h} = \boldsymbol{a}_i^T\boldsymbol{z} \geq 0$. Thus we have

$$
\mathbb{1}_{\{\boldsymbol{a}_i^T\boldsymbol{z}\geq 0\}} \geq \mathbb{1}_{\{\boldsymbol{a}_i^T\boldsymbol{x}\geq 0\}}\mathbb{1}_{\{\boldsymbol{a}_i^T\boldsymbol{h}\geq 0\}}.
$$

Using the above we can conclude that

$$
\begin{aligned}
\nabla\mathcal{L}(\boldsymbol{z})^T(\boldsymbol{z} - \boldsymbol{x}) &\stackrel{(a)}{\geq} \frac{1}{m}\sum_{i=1}^{m}\mathbb{1}_{\{\boldsymbol{a}_i^T\boldsymbol{x}\geq 0\}}\mathbb{1}_{\{\boldsymbol{a}_i^T\boldsymbol{h}\geq 0\}}e^{-\boldsymbol{a}_i^T\boldsymbol{z}}\left(e^{-\boldsymbol{a}_i^T\boldsymbol{x}} - e^{-\boldsymbol{a}_i^T\boldsymbol{z}}\right)(\boldsymbol{a}_i^T\boldsymbol{h}) \\
&\stackrel{(b)}{=} \frac{1}{m}\sum_{i=1}^{m}\mathbb{1}_{\{\boldsymbol{a}_i^T\boldsymbol{x}\geq 0\}}\mathbb{1}_{\{\boldsymbol{a}_i^T\boldsymbol{h}\geq 0\}}e^{-2\boldsymbol{a}_i^T\boldsymbol{x}}e^{-\boldsymbol{a}_i^T\boldsymbol{h}}\left(1 - e^{-\boldsymbol{a}_i^T\boldsymbol{h}}\right)(\boldsymbol{a}_i^T\boldsymbol{h}).
\end{aligned}
$$

In (a) we plug in the indicator inequality, and accordingly remove the now-superfluous ReLU. In (b) we use the $\boldsymbol{h}$ notation, and regroup terms. To continue, we divide both sides by $\|\boldsymbol{h}\|_{\ell_2}^2$ and use the notation $\hat{\boldsymbol{h}} = \frac{\boldsymbol{h}}{\|\boldsymbol{h}\|_{\ell_2}}$.

$$
\frac{1}{\|\boldsymbol{h}\|_{\ell_2}^2}\nabla\mathcal{L}(\boldsymbol{z})^T(\boldsymbol{z} - \boldsymbol{x}) \geq \frac{1}{m}\sum_{i=1}^{m}\mathbb{1}_{\{\boldsymbol{a}_i^T\boldsymbol{x}\geq 0\}}\mathbb{1}_{\{\boldsymbol{a}_i^T\hat{\boldsymbol{h}}\geq 0\}}e^{-2\boldsymbol{a}_i^T\boldsymbol{x}}\frac{e^{-\|\boldsymbol{h}\|_{\ell_2}\boldsymbol{a}_i^T\hat{\boldsymbol{h}}}\left(1 - e^{-\|\boldsymbol{h}\|_{\ell_2}\boldsymbol{a}_i^T\hat{\boldsymbol{h}}}\right)}{\|\boldsymbol{h}\|_{\ell_2}}\boldsymbol{a}_i^T\hat{\boldsymbol{h}}.
$$

To continue note that the function

$$g(x, s) = \frac{e^{-sx}(1 - e^{-sx})}{s}$$

has non-positive derivative as

$$\frac{\partial g}{\partial s} = \frac{e^{-2sx}\left(2sx - e^{sx}(sx + 1) + 1\right)}{s^2} \leq 0$$

for all values of $s$ and $x$. This implies that $g(x, s)$ is a non-increasing function of $s$. Thus, we can conclude that

$$\frac{1}{\|\boldsymbol{h}\|_{\ell_2}^2} \nabla \mathcal{L}(\boldsymbol{z})^T (\boldsymbol{z} - \boldsymbol{x}) \geq \frac{1}{rm} \sum_{i=1}^m \mathbb{1}_{\{\boldsymbol{a}_i^T \boldsymbol{x} \geq 0\}} \mathbb{1}_{\{\boldsymbol{a}_i^T \hat{\boldsymbol{h}} \geq 0\}} e^{-2\boldsymbol{a}_i^T \boldsymbol{x}} e^{-r\boldsymbol{a}_i^T \hat{\boldsymbol{h}}} \left(1 - e^{-r\boldsymbol{a}_i^T \hat{\boldsymbol{h}}}\right) \boldsymbol{a}_i^T \hat{\boldsymbol{h}}.$$

Thus we can focus on lower bounding

$$\frac{1}{rm} \sum_{i=1}^m \mathbb{1}_{\{\boldsymbol{a}_i^T \boldsymbol{x} \geq 0\}} \mathbb{1}_{\{\boldsymbol{a}_i^T \hat{\boldsymbol{h}} \geq 0\}} e^{-2\boldsymbol{a}_i^T \boldsymbol{x}} e^{-r(\boldsymbol{a}_i^T \hat{\boldsymbol{h}})_+} \left(1 - e^{-r(\boldsymbol{a}_i^T \hat{\boldsymbol{h}})_+}\right) (\boldsymbol{a}_i^T \hat{\boldsymbol{h}})_+$$

over the set

$$\left\{\hat{\boldsymbol{h}} \in \mathbb{S}^{n-1} : \frac{\boldsymbol{x}^T \hat{\boldsymbol{h}}}{\|\boldsymbol{x}\|_{\ell_2}} \geq \rho\right\},$$

where we reintroduce superfluous ReLUs around $\boldsymbol{a}_i^T \hat{\boldsymbol{h}}$ as it will be convenient in the next steps. To continue, note that the function $f(h) = e^{-rh}(1 - e^{-rh})h$ has derivative

$$f'(h) = e^{-2rh}\left(-1 - e^{rh}(rh - 1) + 2rh\right).$$

It is easy to verify numerically that for $h \geq 0$ this gradient is maximized around $h_{\max} \approx \frac{0.402673}{r}$, with maximum value $f'(h_{\max}) \approx 0.312334$. Thus, for all $h \geq 0$

$$f'(h) \leq \frac{1}{3}.$$

As a result the function $g(h) = e^{-h_+}(1 - e^{-h_+})h_+$ is a $\frac{1}{3}$-Lipschitz function of $h$. Thus for any $\boldsymbol{h}_1 \in \mathbb{R}^d$ and $\boldsymbol{h}_2 \in \mathbb{R}^d$ we have

$$\left| e^{-(\boldsymbol{a}_i^T \boldsymbol{h}_2)_+} \left(1 - e^{-(\boldsymbol{a}_i^T \boldsymbol{h}_2)_+}\right) (\boldsymbol{a}_i^T \boldsymbol{h}_2)_+ - e^{-(\boldsymbol{a}_i^T \boldsymbol{h}_1)_+} \left(1 - e^{-(\boldsymbol{a}_i^T \boldsymbol{h}_1)_+}\right) (\boldsymbol{a}_i^T \boldsymbol{h}_1)_+ \right| \leq \frac{1}{3} \left| \boldsymbol{a}_i^T (\boldsymbol{h}_2 - \boldsymbol{h}_1) \right|.$$

We now define the random variable $\mathcal{X}_i(\hat{\boldsymbol{h}}) = \mathbb{1}_{\{\boldsymbol{a}_i^T \boldsymbol{x} \geq 0\}} e^{-2\boldsymbol{a}_i^T \boldsymbol{x}} e^{-(\boldsymbol{a}_i^T \hat{\boldsymbol{h}})_+} \left(1 - e^{-(\boldsymbol{a}_i^T \hat{\boldsymbol{h}})_+}\right) (\boldsymbol{a}_i^T \hat{\boldsymbol{h}})_+$
and note that the Lipschitzness of $g$ implies that for any $\boldsymbol{h}_1, \boldsymbol{h}_2 \in \mathbb{R}^d$ we have

$$|\mathcal{X}_i(\boldsymbol{h}_2) - \mathcal{X}_i(\boldsymbol{h}_1)|$$
$$= \mathbb{1}_{\{\boldsymbol{a}_i^T \boldsymbol{x} \geq 0\}} e^{-2\boldsymbol{a}_i^T \boldsymbol{x}} \left| e^{-(\boldsymbol{a}_i^T \boldsymbol{h}_2)_+} \left(1 - e^{-(\boldsymbol{a}_i^T \boldsymbol{h}_2)_+}\right) (\boldsymbol{a}_i^T \boldsymbol{h}_2)_+ - e^{-(\boldsymbol{a}_i^T \boldsymbol{h}_1)_+} \left(1 - e^{-(\boldsymbol{a}_i^T \boldsymbol{h}_1)_+}\right) (\boldsymbol{a}_i^T \boldsymbol{h}_1)_+ \right|$$
$$\leq \frac{1}{3} \left| \boldsymbol{a}_i^T (\boldsymbol{h}_2 - \boldsymbol{h}_1) \right|.$$

Since $\boldsymbol{a}_i^T (\boldsymbol{h}_2 - \boldsymbol{h}_1)$ is a sub-Gaussian random variable with sub-Gaussian norm on the order of $\|\boldsymbol{h}_2 - \boldsymbol{h}_1\|_{\ell_2}$, we have that

$$\|\mathcal{X}_i(\boldsymbol{h}_2) - \mathcal{X}_i(\boldsymbol{h}_1)\|_{\psi_2} \leq \frac{c}{2} \|\boldsymbol{h}_2 - \boldsymbol{h}_1\|_{\ell_2}$$

for some constant $c$. We use $c$ to denote any universal constant; note that this constant may vary between different lines. Thus using the centering rule for sub-Gaussian random variables, the centered processes $\bar{\mathcal{X}}_i(\boldsymbol{h}) := \mathcal{X}_i(\boldsymbol{h}) - \mathbb{E}[\mathcal{X}_i(\boldsymbol{h})]$ obey

$$\|\bar{\mathcal{X}}_i(\boldsymbol{h}_2) - \bar{\mathcal{X}}_i(\boldsymbol{h}_1)\|_{\psi_2} \leq c \|\boldsymbol{h}_2 - \boldsymbol{h}_1\|_{\ell_2}.$$

Using the rotational invariance property of sub-Gaussian random variables, this implies that the stochastic process

$$\mathcal{X}(\boldsymbol{h}) := \frac{1}{m} \sum_{i=1}^{m} \mathcal{X}_i(\boldsymbol{h}) - \mathbb{E}[\mathcal{X}_i(\boldsymbol{h})]$$

has sub-Gaussian increments. That is,

$$\|\mathcal{X}(\boldsymbol{h}_2) - \mathcal{X}(\boldsymbol{h}_1)\|_{\psi_2} \leq \frac{c}{\sqrt{m}} \|\boldsymbol{h}_2 - \boldsymbol{h}_1\|_{\ell_2}.$$

Thus using Exercise 8.6.5 of Vershynin (2018) we can conclude that

$$\sup_{\|\hat{\boldsymbol{h}}\|_{\ell_2}=1} \frac{\left|\mathcal{X}(\hat{\boldsymbol{h}})\right|}{r} \leq \frac{c}{\sqrt{mr}} \left(\sqrt{n} + u\right)$$

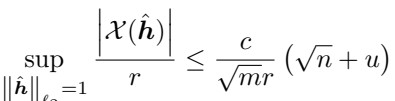



Figure 5: Lower bound on the expected correlation in case 1.

holds with probability at least $1 - 2e^{-u^2}$. Thus, we conclude that for all $\boldsymbol{h}$ obeying $\|\boldsymbol{h}\|_{\ell_2} \leq r$ we have

$$\frac{1}{\|\boldsymbol{h}\|_{\ell_2}^2} \nabla \mathcal{L}(\boldsymbol{z})^T (\boldsymbol{z} - \boldsymbol{x}) \geq \frac{1}{r} \mathbb{E}\left[\mathbb{1}_{\{\boldsymbol{a}^T\boldsymbol{x}\geq 0\}} \mathbb{1}_{\{\boldsymbol{a}^T\hat{\boldsymbol{h}}\geq 0\}} e^{-2\boldsymbol{a}^T\boldsymbol{x}} e^{-r\boldsymbol{a}_i^T\hat{\boldsymbol{h}}} \left(1 - e^{-r\boldsymbol{a}_i^T\hat{\boldsymbol{h}}}\right) \boldsymbol{a}_i^T\hat{\boldsymbol{h}}\right] - \frac{c}{r}\frac{\sqrt{n}}{\sqrt{m}}$$

with probability at least $1 - 2e^{-n}$, for a constant $c$.

We can estimate and lower bound the expectation above using a numerical average over many (50000) two-dimensional Gaussian samples, with the two dimensions corresponding to $\boldsymbol{a}^T\hat{\boldsymbol{h}}$ and $\frac{\boldsymbol{a}^T\boldsymbol{x}}{\|\boldsymbol{x}\|_{\ell_2}}$, minimizing over all correlations between $\boldsymbol{x}$ and $\boldsymbol{h}$ at least $\rho$ (*i.e.* all correlations in this case). We arrive at the following lower bound, for $r = \frac{1}{4} \|\boldsymbol{x}\|_{\ell_2}$ and $\rho = -0.6$.

$$\frac{1}{r} \mathbb{E}\left[\mathbb{1}_{\{\boldsymbol{a}^T\boldsymbol{x}\geq 0\}} \mathbb{1}_{\{\boldsymbol{a}^T\hat{\boldsymbol{h}}\geq 0\}} e^{-2\boldsymbol{a}^T\boldsymbol{x}} e^{-r\boldsymbol{a}_i^T\hat{\boldsymbol{h}}} \left(1 - e^{-r\boldsymbol{a}_i^T\hat{\boldsymbol{h}}}\right) \boldsymbol{a}_i^T\hat{\boldsymbol{h}}\right] \geq e^{-\sqrt{10\|\boldsymbol{x}\|_{\ell_2}+7}}.$$

This bound is illustrated in a "proof by picture" in Figure 5.

### B.4   CORRELATION CONCENTRATION: CASE 2

In this case we will focus on controlling the correlation inequality in the region where

$$\left\{\boldsymbol{h} \in \mathbb{R}^n : \|\boldsymbol{h}\|_{\ell_2} \leq r \quad \text{and} \quad \frac{\boldsymbol{x}^T\boldsymbol{h}}{\|\boldsymbol{x}\|_{\ell_2}\|\boldsymbol{h}\|_{\ell_2}} \leq \rho\right\}.$$

Consider the correlation

$$\nabla \mathcal{L}(\boldsymbol{z})^T(\boldsymbol{z}-\boldsymbol{x}) = \frac{1}{m}\sum_{i=1}^{m} \mathbb{1}_{\{\boldsymbol{a}_i^T\boldsymbol{z}\geq 0\}} e^{-\boldsymbol{a}_i^T\boldsymbol{z}} \left(e^{-(\boldsymbol{a}_i^T\boldsymbol{x})_+} - e^{-\boldsymbol{a}_i^T\boldsymbol{z}}\right)(\boldsymbol{a}_i^T\boldsymbol{h})$$

$$\overset{(a)}{\geq} \frac{1}{m}\sum_{i=1}^{m} \mathbb{1}_{\{\boldsymbol{a}_i^T\boldsymbol{x}\geq 0\}} \mathbb{1}_{\{\boldsymbol{a}_i^T\boldsymbol{z}\geq 0\}} e^{-\boldsymbol{a}_i^T\boldsymbol{z}} \left(e^{-\boldsymbol{a}_i^T\boldsymbol{x}} - e^{-\boldsymbol{a}_i^T\boldsymbol{z}}\right)(\boldsymbol{a}_i^T\boldsymbol{h})$$

$$\overset{(b)}{=} \frac{1}{m}\sum_{i=1}^{m} \mathbb{1}_{\{\boldsymbol{a}_i^T\boldsymbol{x}\geq 0\}} \mathbb{1}_{\{\boldsymbol{a}_i^T\boldsymbol{h}\geq -\boldsymbol{a}_i^T\boldsymbol{x}\}} e^{-2\boldsymbol{a}_i^T\boldsymbol{x}} e^{-\boldsymbol{a}_i^T\boldsymbol{h}} \left(1 - e^{-\boldsymbol{a}_i^T\boldsymbol{h}}\right)(\boldsymbol{a}_i^T\boldsymbol{h})$$

$$\overset{(c)}{\geq} \frac{1}{m}\sum_{i=1}^{m} \mathbb{1}_{\{\boldsymbol{a}_i^T\boldsymbol{x}\geq 0\}} \mathbb{1}_{\{0\geq \boldsymbol{a}_i^T\boldsymbol{h}\geq -\boldsymbol{a}_i^T\boldsymbol{x}\}} e^{-2\boldsymbol{a}_i^T\boldsymbol{x}} e^{-\boldsymbol{a}_i^T\boldsymbol{h}} \left(1 - e^{-\boldsymbol{a}_i^T\boldsymbol{h}}\right)(\boldsymbol{a}_i^T\boldsymbol{h}).$$

In (a) we provide a lower bound by introducing an additional indicator function on $\boldsymbol{a}_i^T\boldsymbol{x}$, which allows us to remove the ReLU. In (b) we use the notation $\boldsymbol{h} := \boldsymbol{z} - \boldsymbol{x}$, and combine terms. In (c) we

again provide a lower bound by adding an indicator to restrict $\boldsymbol{a}_i^T \boldsymbol{h} \leq 0$. By flipping the sign of $\boldsymbol{h}$ we can alternatively lower bound

$$\frac{1}{m} \sum_{i=1}^m \mathbb{1}_{\{\boldsymbol{a}_i^T \boldsymbol{x} \geq 0\}} \mathbb{1}_{\{0 \leq \boldsymbol{a}_i^T \boldsymbol{h} \leq \boldsymbol{a}_i^T \boldsymbol{x}\}} e^{-2\boldsymbol{a}_i^T \boldsymbol{x}} e^{\boldsymbol{a}_i^T \boldsymbol{h}} \left( e^{\boldsymbol{a}_i^T \boldsymbol{h}} - 1 \right) (\boldsymbol{a}_i^T \boldsymbol{h})$$

over the set

$$\left\{ \boldsymbol{h} \in \mathbb{R}^n : \|\boldsymbol{h}\|_{\ell_2} \leq r \quad \text{and} \quad \frac{\boldsymbol{x}^T \boldsymbol{h}}{\|\boldsymbol{x}\|_{\ell_2} \|\boldsymbol{h}\|_{\ell_2}} \geq -\rho \right\}.$$

To this aim note that for $s \geq 0$ we have $e^s \geq 1$ and $(e^s - 1) s \geq s^2$. Thus,

$$\nabla \mathcal{L}(\boldsymbol{z})^T (\boldsymbol{z} - \boldsymbol{x}) \geq \frac{1}{m} \sum_{i=1}^m \mathbb{1}_{\{\boldsymbol{a}_i^T \boldsymbol{x} \geq 0\}} \mathbb{1}_{\{0 \leq \boldsymbol{a}_i^T \boldsymbol{h} \leq \boldsymbol{a}_i^T \boldsymbol{x}\}} e^{-2\boldsymbol{a}_i^T \boldsymbol{x}} (\boldsymbol{a}_i^T \boldsymbol{h})^2.$$

To continue, we introduce the notation $\hat{\boldsymbol{h}} = \frac{\boldsymbol{h}}{\|\boldsymbol{h}\|_{\ell_2}}$, and divide both sides by $\|\boldsymbol{h}\|_{\ell_2}^2$. Thus we have

$$\frac{1}{\|\boldsymbol{h}\|_{\ell_2}^2} \nabla \mathcal{L}(\boldsymbol{z})^T (\boldsymbol{z} - \boldsymbol{x}) \geq \frac{1}{m} \sum_{i=1}^m \mathbb{1}_{\{\boldsymbol{a}_i^T \boldsymbol{x} \geq 0\}} \mathbb{1}_{\{0 \leq \boldsymbol{a}_i^T \hat{\boldsymbol{h}} \leq \boldsymbol{a}_i^T \boldsymbol{x}/\|\boldsymbol{h}\|_{\ell_2}\}} e^{-2\boldsymbol{a}_i^T \boldsymbol{x}} \left( \boldsymbol{a}_i^T \hat{\boldsymbol{h}} \right)^2$$

$$\geq \frac{1}{m} \sum_{i=1}^m \mathbb{1}_{\{\boldsymbol{a}_i^T \boldsymbol{x} \geq 0\}} \mathbb{1}_{\{0 \leq \boldsymbol{a}_i^T \hat{\boldsymbol{h}} \leq \boldsymbol{a}_i^T \boldsymbol{x}/r\}} e^{-2\boldsymbol{a}_i^T \boldsymbol{x}} \left( \boldsymbol{a}_i^T \hat{\boldsymbol{h}} \right)^2.$$

Thus it suffices to lower bound

$$\frac{1}{m} \sum_{i=1}^m \mathbb{1}_{\{\boldsymbol{a}_i^T \boldsymbol{x} \geq 0\}} \mathbb{1}_{\left\{0 \leq \boldsymbol{a}_i^T \hat{\boldsymbol{h}} \leq \frac{\boldsymbol{a}_i^T \boldsymbol{x}}{r}\right\}} e^{-2\boldsymbol{a}_i^T \boldsymbol{x}} (\boldsymbol{a}_i^T \hat{\boldsymbol{h}})^2$$

over

$$\left\{ \hat{\boldsymbol{h}} \in \mathbb{S}^{n-1} : \frac{\boldsymbol{x}^T \hat{\boldsymbol{h}}}{\|\boldsymbol{x}\|_{\ell_2}} \geq -\rho \right\}.$$

To continue using Jensen's inequality we have

$$\frac{1}{\|\boldsymbol{h}\|_{\ell_2}^2} \nabla \mathcal{L}(\boldsymbol{z})^T (\boldsymbol{z} - \boldsymbol{x}) \geq \left( \frac{1}{m} \sum_{i=1}^m \mathbb{1}_{\{\boldsymbol{a}_i^T \boldsymbol{x} \geq 0\}} \mathbb{1}_{\left\{0 \leq \boldsymbol{a}_i^T \hat{\boldsymbol{h}} \leq \frac{\boldsymbol{a}_i^T \boldsymbol{x}}{r}\right\}} e^{-\boldsymbol{a}_i^T \boldsymbol{x}} \boldsymbol{a}_i^T \hat{\boldsymbol{h}} \right)^2$$

$$\geq \left( \frac{1}{m} \sum_{i=1}^m \mathbb{1}_{\{\boldsymbol{a}_i^T \boldsymbol{x} \geq 0\}} \mathcal{S} \left( \boldsymbol{a}_i^T \hat{\boldsymbol{h}}; \frac{\boldsymbol{a}_i^T \boldsymbol{x}}{r} \right) e^{-\boldsymbol{a}_i^T \boldsymbol{x}} \right)^2$$

where we have defined the function

$$\mathcal{S}(v; w) = \begin{cases} 0 & v < 0 \\ v & 0 \leq v \leq \frac{w}{2} \\ w - v & \frac{w}{2} \leq v \leq w \\ 0 & v \geq w \end{cases}$$

which is a 1-Lipschitz function of $v$. We now define the random variable $\mathcal{X}_i(\hat{\boldsymbol{h}}) = \mathbb{1}_{\{\boldsymbol{a}_i^T \boldsymbol{x} \geq 0\}} \mathcal{S} \left( \boldsymbol{a}_i^T \hat{\boldsymbol{h}}; \frac{\boldsymbol{a}_i^T \boldsymbol{x}}{r} \right) e^{-\boldsymbol{a}_i^T \boldsymbol{x}}$ and note that the Lipschitzness of $\mathcal{S}$ implies that for any $\boldsymbol{h}_1, \boldsymbol{h}_2 \in \mathbb{R}^d$ we have

$$|\mathcal{X}_i(\boldsymbol{h}_2) - \mathcal{X}_i(\boldsymbol{h}_1)| \leq \left| \boldsymbol{a}_i^T (\boldsymbol{h}_2 - \boldsymbol{h}_1) \right|.$$

Since $\boldsymbol{a}_i^T (\boldsymbol{h}_2 - \boldsymbol{h}_1)$ is a sub-Gaussian random variable with sub-Gaussian norm on the order of $\|\boldsymbol{h}_2 - \boldsymbol{h}_1\|_{\ell_2}$, we have that

$$\|\mathcal{X}_i(\boldsymbol{h}_2) - \mathcal{X}_i(\boldsymbol{h}_1)\|_{\psi_2} \leq \frac{c}{2} \|\boldsymbol{h}_2 - \boldsymbol{h}_1\|_{\ell_2}$$

for some constant $c$. We use $c$ to denote any universal constant; note that this constant may vary between different lines. Using the centering rule for sub-Gaussian random variables, the centered processes $\bar{\mathcal{X}}_i(\boldsymbol{h}) := \mathcal{X}_i(\boldsymbol{h}) - \mathbb{E}[\mathcal{X}_i(\boldsymbol{h})]$ obey

$$\left\| \bar{\mathcal{X}}_i(\boldsymbol{h}_2) - \bar{\mathcal{X}}_i(\boldsymbol{h}_1) \right\|_{\psi_2} \le c \left\| \boldsymbol{h}_2 - \boldsymbol{h}_1 \right\|_{\ell_2}.$$

Using the rotational invariance property of sub-Gaussian random variables, this implies that the stochastic process

$$\mathcal{X}(\boldsymbol{h}) := \frac{1}{m} \sum_{i=1}^{m} \mathcal{X}_i(\boldsymbol{h}) - \mathbb{E}[\mathcal{X}_i(\boldsymbol{h})]$$

has sub-Gaussian increments. That is,

$$\left\| \mathcal{X}(\boldsymbol{h}_2) - \mathcal{X}(\boldsymbol{h}_1) \right\|_{\psi_2} \le \frac{c}{\sqrt{m}} \left\| \boldsymbol{h}_2 - \boldsymbol{h}_1 \right\|_{\ell_2}.$$

Thus using Exercise 8.6.5 of Vershynin (2018) we can conclude that

$$\sup_{\|\hat{\boldsymbol{h}}\|_{\ell_2}=1, \cos^{-1}\left(\frac{\boldsymbol{x}^T \hat{\boldsymbol{h}}}{\|\boldsymbol{x}\|_{\ell_2}}\right) \le \delta} \left| \mathcal{X}(\hat{\boldsymbol{h}}) \right| \le \frac{c}{\sqrt{m}} \left( \sqrt{n} e^{-\frac{n \cos^2(\delta)}{2}} + u \right)$$

holds with probability at least $1 - 2e^{-u^2}$. In the last line we used the fact that the surface area of a spherical cap with distance at least $\epsilon$ away from the center is bounded by $e^{-n\frac{\epsilon^2}{2}}$. By using $u = \sqrt{n}$, this implies that

$$\frac{1}{\|\boldsymbol{h}\|_{\ell_2}^2} \nabla \mathcal{L}(\boldsymbol{z})^T (\boldsymbol{z} - \boldsymbol{x}) \ge \left( \mathbb{E}\left[ \mathbb{1}_{\{\boldsymbol{a}^T \boldsymbol{x} \ge 0\}} \mathcal{S}\left( \boldsymbol{a}^T \hat{\boldsymbol{h}}; \frac{\boldsymbol{a}_i^T \boldsymbol{x}}{r} \right) e^{-\boldsymbol{a}^T \boldsymbol{x}} \right] - c \frac{\sqrt{n}}{\sqrt{m}} \right)^2$$

holds with probability at least $1 - 2e^{-n}$, for a constant $c$.

We can estimate and lower bound the expectation above using a numerical average over many (50000) two-dimensional Gaussian samples, with the two dimensions corresponding to $\boldsymbol{a}^T \hat{\boldsymbol{h}}$ and $\frac{\boldsymbol{a}^T \boldsymbol{x}}{\|\boldsymbol{x}\|_{\ell_2}}$, minimizing over all correlations between $\boldsymbol{x}$ and $\boldsymbol{h}$ at most $\rho$ (*i.e.* all correlations in this case). We arrive at the following lower bound, for $r = \frac{1}{4} \|\boldsymbol{x}\|_{\ell_2}$ and $\rho = -0.6$.

$$\mathbb{E}\left[ \mathbb{1}_{\{\boldsymbol{a}^T \boldsymbol{x} \ge 0\}} \mathcal{S}\left( \boldsymbol{a}^T \hat{\boldsymbol{h}}; \frac{\boldsymbol{a}_i^T \boldsymbol{x}}{r} \right) e^{-\boldsymbol{a}^T \boldsymbol{x}} \right]^2 \ge e^{-(5\|\boldsymbol{x}\|_{\ell_2} + 2)}.$$

This bound is illustrated in a "proof by picture" in Figure 6.

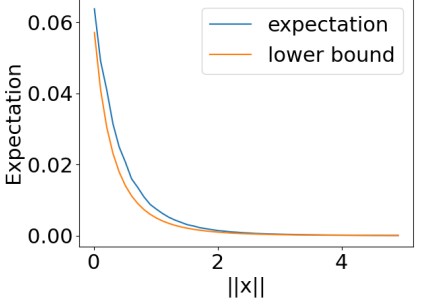

Figure 6: Lower bound on the expected correlation in case 2.

## B.5 BOUNDING THE GRADIENT NORM

Consider the gradient and note that

$$\left\| \nabla \mathcal{L}(\boldsymbol{z}) \right\|_{\ell_2} = \sup_{\boldsymbol{u} \in \mathbb{S}^{n-1}} \frac{1}{m} \sum_{i=1}^{m} \mathbb{1}_{\{\boldsymbol{a}_i^T \boldsymbol{z} \ge 0\}} e^{-(\boldsymbol{a}_i^T \boldsymbol{z})_+} \left( e^{-(\boldsymbol{a}_i^T \boldsymbol{x})_+} - e^{-(\boldsymbol{a}_i^T \boldsymbol{z})_+} \right) (\boldsymbol{a}_i^T \boldsymbol{u}),$$

where $\mathbb{S}^{n-1}$ denotes the set of all real $n$-dimensional unit-norm vectors. To continue, we use the Cauchy-Schwarz inequality:

$$
\begin{aligned}
\|\nabla\mathcal{L}(\boldsymbol{z})\|_{\ell_2} &\leq \sqrt{\frac{1}{m}\sum_{i=1}^m \mathbb{1}_{\{\boldsymbol{a}_i^T\boldsymbol{z}\geq 0\}} e^{-2(\boldsymbol{a}_i^T\boldsymbol{z})_+}\left(e^{-(\boldsymbol{a}_i^T\boldsymbol{x})_+} - e^{-(\boldsymbol{a}_i^T\boldsymbol{z})_+}\right)^2} \sqrt{\sup_{\boldsymbol{u}\in\mathbb{S}^{n-1}}\frac{1}{m}\sum_{i=1}^m (\boldsymbol{a}_i^T\boldsymbol{u})^2} \\
&= \sqrt{\frac{1}{m}\sum_{i=1}^m \mathbb{1}_{\{\boldsymbol{a}_i^T\boldsymbol{z}\geq 0\}} e^{-2(\boldsymbol{a}_i^T\boldsymbol{z})_+}\left(e^{-(\boldsymbol{a}_i^T\boldsymbol{x})_+} - e^{-(\boldsymbol{a}_i^T\boldsymbol{z})_+}\right)^2} \frac{\|\boldsymbol{A}\|}{\sqrt{m}} \\
&\leq \sqrt{\frac{1}{m}\sum_{i=1}^m \left(e^{-(\boldsymbol{a}_i^T\boldsymbol{x})_+} - e^{-(\boldsymbol{a}_i^T\boldsymbol{z})_+}\right)^2} \frac{\|\boldsymbol{A}\|}{\sqrt{m}} \\
&\overset{(a)}{\leq} \sqrt{\frac{1}{m}\sum_{i=1}^m \left(\boldsymbol{a}_i^T(\boldsymbol{z}-\boldsymbol{x})\right)^2} \frac{\|\boldsymbol{A}\|}{\sqrt{m}} \\
&\leq \frac{\|\boldsymbol{A}\|^2}{m}\|\boldsymbol{z}-\boldsymbol{x}\|_{\ell_2},
\end{aligned}
$$

where $\|\boldsymbol{A}\|$ is the operator norm of a matrix comprised by stacking the vectors $\boldsymbol{a}_i$, and in (a) we used the fact that the function $f(z) = e^{-(z)_+}$ is 1-Lipschitz. Finally, using the fact that $\|\boldsymbol{A}\| \leq 2(\sqrt{m} + \sqrt{n})$ with probability at least $1 - e^{-0.5(m+n)}$, we conclude that

$$
\|\nabla\mathcal{L}(\boldsymbol{z})\|_{\ell_2} \leq 8\left(1 + \frac{n}{m}\right)\|\boldsymbol{z}-\boldsymbol{x}\|_{\ell_2} := L\|\boldsymbol{z}-\boldsymbol{x}\|_{\ell_2}
$$

with probability at least $1 - e^{-0.5(m+n)}$, where $L$ is a constant since we have the number of measurements at least a constant times the number of unknowns. This completes the proof of smoothness of the gradient towards the global optimum.

## C  PROOF OF THEOREM 2

The general strategy of the proof is similar to Theorem 1 but requires delicate modifications in each step. Concretely, we have the following four steps.

**Step I: First iteration: Welcome to the neighborhood.**
In the first step we show that the first iteration obeys

$$
\|\boldsymbol{z}_1 - \boldsymbol{x}\|_{\ell_2} \leq \frac{1}{4}\|\boldsymbol{x}\|_{\ell_2}
$$

with high probability as long as $m \geq cm_0$. We prove this in subsection C.1.

**Step II: Local pseudoconvexity.**
In this step we prove that the loss function is locally strongly pseudoconvex. Specifically we show that for all $\boldsymbol{z} \in \mathcal{K}$ that also belong to the local neighborhoood $\mathcal{N}(\boldsymbol{x}) := \{\boldsymbol{z}\in\mathbb{R}^n : \|\boldsymbol{z}-\boldsymbol{x}\|_{\ell_2} \leq \frac{1}{4}\|\boldsymbol{x}\|_{\ell_2}\}$ we have

$$
\langle\nabla\mathcal{L}(\boldsymbol{z}), \boldsymbol{z}-\boldsymbol{x}\rangle \geq \alpha\|\boldsymbol{z}-x\|_{\ell_2}^2
$$

with high probability as long as

$$
m \geq \frac{c_1 e^{c_2\|\boldsymbol{x}\|_{\ell_2}}}{\|\boldsymbol{x}\|_{\ell_2}^2}m_0.
$$

Here, the value of $\alpha$ is the same as in Theorem 1 (see Appendix B.1). We prove this in subsection C.2 by again considering two cases.

**Step III: Local smoothness.**
We also use the fact that the loss function is locally smooth, that is,

$$
\|\nabla\mathcal{L}(\boldsymbol{z})\|_{\ell_2} \leq 8\left(1 + \frac{n}{m}\right)\|\boldsymbol{z}-\boldsymbol{x}\|_{\ell_2} := L\|\boldsymbol{z}-\boldsymbol{x}\|_{\ell_2}
$$

holds with probability at least $1 - e^{-0.5(m+n)}$ per Section B.5.

**Step IV: Completing the proof via combining steps I-III.**
In this step we show how to combine the previous steps to complete the proof of the theorem. First, note that by the first step the first iteration will belong to the local neighborhood $\mathcal{N}(\boldsymbol{x})$ and thus belongs to the set $\mathcal{K} \cap \mathcal{N}(\boldsymbol{x})$. Next, note that

$$
\begin{aligned}
\|\boldsymbol{z}_{t+1} - \boldsymbol{x}\|_{\ell_2} &= \|\mathcal{P}_{\mathcal{K}}(\boldsymbol{z}_t - \mu_{t+1}\nabla\mathcal{L}(\boldsymbol{z}_t)) - \boldsymbol{x}\|_{\ell_2} \\
&= \|\mathcal{P}_{\mathcal{D}}(\boldsymbol{h}_t - \mu_{t+1}\nabla\mathcal{L}(\boldsymbol{z}_t))\|_{\ell_2} \\
&\leq \|\mathcal{P}_{\mathcal{C}}(\boldsymbol{h}_t - \mu_{t+1}\nabla\mathcal{L}(\boldsymbol{z}_t))\|_{\ell_2} \\
&\leq \|\boldsymbol{h}_t - \mu_{t+1}\nabla\mathcal{L}(\boldsymbol{z}_t)\|_{\ell_2}.
\end{aligned}
$$

Squaring both sides and using the local psudoconvexity inequality from Step II we conclude that

$$
\begin{aligned}
\|\boldsymbol{z}_{t+1} - \boldsymbol{x}\|_{\ell_2}^2 &\leq \|\boldsymbol{h}_t - \mu_{t+1}\nabla\mathcal{L}(\boldsymbol{z}_t)\|_{\ell_2}^2 \\
&= \|\boldsymbol{h}_t\|_{\ell_2}^2 - 2\mu_{t+1}\langle \boldsymbol{h}_t, \nabla\mathcal{L}(\boldsymbol{z}_t)\rangle + \mu_{t+1}^2 \|\nabla\mathcal{L}(\boldsymbol{z}_t)\|_{\ell_2}^2 \\
&\leq \|\boldsymbol{h}_t\|_{\ell_2}^2 - 2\mu_{t+1}\alpha \|\boldsymbol{h}_t\|_{\ell_2}^2 + \mu_{t+1}^2 \|\nabla\mathcal{L}(\boldsymbol{z}_t)\|_{\ell_2}^2.
\end{aligned}
$$

Next we use the local smoothness from Step III to conclude that

$$
\begin{aligned}
\|\boldsymbol{z}_{t+1} - \boldsymbol{x}\|_{\ell_2}^2 &\leq \|\boldsymbol{h}_t\|_{\ell_2}^2 - 2\mu_{t+1}\alpha \|\boldsymbol{h}_t\|_{\ell_2}^2 + \mu_{t+1}^2 \|\nabla\mathcal{L}(\boldsymbol{z}_t)\|_{\ell_2}^2 \\
&\leq \|\boldsymbol{h}_t\|_{\ell_2}^2 - 2\mu_{t+1}\alpha \|\boldsymbol{h}_t\|_{\ell_2}^2 + \mu_{t+1}^2 L^2 \|\boldsymbol{h}_t\|_{\ell_2}^2 \\
&= (1 - \mu_{t+1}\alpha)\|\boldsymbol{h}_t\|_{\ell_2}^2 \\
&= (1 - \mu_{t+1}\alpha)\|\boldsymbol{z}_t - \boldsymbol{x}\|_{\ell_2}^2,
\end{aligned}
$$

where in the last line we used the fact that $\mu_{t+1} \leq \frac{\alpha}{L^2}$, completing the proof.

## C.1 PROOF OF STEP I

The beginning of this proof is the same as the unregularized version where we note that

$$
\begin{aligned}
\|\boldsymbol{z}_1 - \boldsymbol{x}\|_{\ell_2} &= \left\|\mathcal{P}_{\mathcal{K}}\left(\frac{\mu_1}{2m}\sum_{i=1}^{m}\hat{g}_i\frac{\boldsymbol{x}}{\|\boldsymbol{x}\|_{\ell_2}} + \frac{\mu_1}{2m}\sqrt{\sum_{i=1}^{m}\tilde{g}_i\boldsymbol{a}_\perp}\right) - \boldsymbol{x}\right\|_{\ell_2} \\
&= \left\|\mathcal{P}_{\mathcal{D}}\left(\frac{\mu_1}{2m}\sum_{i=1}^{m}\hat{g}_i\frac{\boldsymbol{x}}{\|\boldsymbol{x}\|_{\ell_2}} + \frac{\mu_1}{2m}\sqrt{\sum_{i=1}^{m}\tilde{g}_i\boldsymbol{a}_\perp - \boldsymbol{x}}\right)\right\|_{\ell_2} \\
&= \left\|\mathcal{P}_{\mathcal{D}}\left(\frac{\mu_1}{2m}\sum_{i=1}^{m}(\hat{g}_i - \mathbb{E}[\hat{g}])\frac{\boldsymbol{x}}{\|\boldsymbol{x}\|_{\ell_2}} + \frac{\mu_1}{2m}\sqrt{\sum_{i=1}^{m}\tilde{g}_i\boldsymbol{a}_\perp}\right)\right\|_{\ell_2} \\
&\leq \left\|\mathcal{P}_{\mathcal{C}}\left(\frac{\mu_1}{2m}\sum_{i=1}^{m}(\hat{g}_i - \mathbb{E}[\hat{g}])\frac{\boldsymbol{x}}{\|\boldsymbol{x}\|_{\ell_2}} + \frac{\mu_1}{2m}\sqrt{\sum_{i=1}^{m}\tilde{g}_i\boldsymbol{a}_\perp}\right)\right\|_{\ell_2} \\
&\leq \sup_{\boldsymbol{v}\in\mathcal{C}\cap\mathbb{S}^{n-1}}\boldsymbol{v}^T\left(\frac{\mu_1}{2m}\sum_{i=1}^{m}(\hat{g}_i - \mathbb{E}[\hat{g}])\frac{\boldsymbol{x}}{\|\boldsymbol{x}\|_{\ell_2}} + \frac{\mu_1}{2m}\sqrt{\sum_{i=1}^{m}\tilde{g}_i\boldsymbol{a}_\perp}\right) \\
&\leq \frac{\mu_1}{2}\left|\frac{1}{m}\sum_{i=1}^{m}(\hat{g}_i - \mathbb{E}[\hat{g}])\right| + \frac{\mu_1}{2\sqrt{m}}\sup_{\boldsymbol{v}\in\mathcal{C}\cap\mathbb{S}^{n-1}}\boldsymbol{v}^T\boldsymbol{a}_\perp.
\end{aligned}
$$

Now similar to the unregularized case we have

$$\frac{\mu_1}{2m}\left|\sum_{i=1}^{m}(\hat{g}_i - \mathbb{E}[\hat{g}])\right| + \frac{\mu_1}{2\sqrt{m}}\sup_{\boldsymbol{v}\in\mathcal{C}\cap\mathbb{S}^{n-1}}\boldsymbol{v}^T\boldsymbol{a}_\perp \le \frac{\mu_1}{2}\left(\frac{s}{m} + (1+t)\sqrt{\frac{m_0}{m}}\right)$$

$$= 2\exp\left(-\frac{\|\boldsymbol{x}\|_{\ell_2}^2}{2}\right)\frac{1}{\operatorname{erfc}\left(\frac{\|\boldsymbol{x}\|_{\ell_2}}{\sqrt{2}}\right)}\left(\frac{s}{m} + (1+t)\sqrt{\frac{m_0}{m}}\right)$$

$$\stackrel{(a)}{=} 2\exp\left(-\frac{\|\boldsymbol{x}\|_{\ell_2}^2}{2}\right)\frac{1}{\operatorname{erfc}\left(\frac{\|\boldsymbol{x}\|_{\ell_2}}{\sqrt{2}}\right)}\left(s + (1+t)\sqrt{\frac{m_0}{m}}\right)$$

with probability at least $1 - 2e^{-\frac{s^2 m}{2}} - e^{-ct^2 m_0}$, where in (a) we change variables and replace $s$ with $sm$, and $c$ is a constant. If we choose $s = t = 1$, this simplifies to

$$\|\boldsymbol{z}_1 - \boldsymbol{x}\|_{\ell_2} \le 2\exp\left(-\frac{\|\boldsymbol{x}\|_{\ell_2}^2}{2}\right)\frac{1}{\operatorname{erfc}\left(\frac{\|\boldsymbol{x}\|_{\ell_2}}{\sqrt{2}}\right)}\left(1 + 2\sqrt{\frac{m_0}{m}}\right)$$

with probability at least $1 - 2e^{-\frac{m}{2}} - e^{-cm_0}$, for a constant $c$. For our first step to lie within a distance of $\frac{1}{4}\|\boldsymbol{x}\|_{\ell_2}$, we need the number of measurements to satisfy

$$m \ge \frac{m_0}{\left(\frac{\|\boldsymbol{x}\|_{\ell_2}}{16}\exp\left(\frac{\|\boldsymbol{x}\|_{\ell_2}^2}{2}\right)\operatorname{erfc}\left(\frac{\|\boldsymbol{x}\|_{\ell_2}}{\sqrt{2}}\right) - \frac{1}{2}\right)^2}.$$

The denominator in this expression is lower bounded by $0.2$ for all $\|\boldsymbol{x}\|_{\ell_2}$, so we can also guarantee the first step concentration to this neighborhood using $m \ge 5m_0$ measurements.

## C.2 PROOF OF STEP II

The proof of this step is virtually identical to that of the unregularized case. The only difference is that when we apply Exercise 8.6.5 of Vershynin (2018) $n$ is replaced with $m_0$ (indeed this exercise is stated with $m_0$). As a result in the two cases we conclude

**Case I:** In this case using the above yields

$$\frac{1}{\|\boldsymbol{h}\|_{\ell_2}^2}\nabla\mathcal{L}(\boldsymbol{z})^T(\boldsymbol{z} - \boldsymbol{x}) \ge \frac{1}{r}\mathbb{E}\left[\mathbb{1}_{\{\boldsymbol{a}^T\boldsymbol{x}\ge 0\}}\mathbb{1}_{\{\boldsymbol{a}^T\hat{\boldsymbol{h}}\ge 0\}}e^{-2\boldsymbol{a}^T\boldsymbol{x}}e^{-r\boldsymbol{a}_i^T\hat{\boldsymbol{h}}}\left(1 - e^{-r\boldsymbol{a}_i^T\hat{\boldsymbol{h}}}\right)\boldsymbol{a}_i^T\hat{\boldsymbol{h}}\right] - \frac{c}{r}\frac{\sqrt{m_0}}{\sqrt{m}}$$

holds with probability at least $1 - 2e^{-m_0}$, for a constant $c$.

**Case II:** In this case using the above yields

$$\frac{1}{\|\boldsymbol{h}\|_{\ell_2}^2}\nabla\mathcal{L}(\boldsymbol{z})^T(\boldsymbol{z} - \boldsymbol{x}) \ge \left(\mathbb{E}\left[\mathbb{1}_{\{\boldsymbol{a}^T\boldsymbol{x}\ge 0\}}\mathcal{S}\left(\boldsymbol{a}^T\hat{\boldsymbol{h}};\frac{\boldsymbol{a}_i^T\boldsymbol{x}}{r}\right)e^{-\boldsymbol{a}^T\boldsymbol{x}}\right] - c\frac{\sqrt{m_0}}{\sqrt{m}}\right)^2$$

holds with probability at least $1 - 2e^{-m_0}$, for a constant $c$.

Thus the remainder of the proof is identical and the only needed change in this entire step is to replace $n$ with $m_0$.

