# OpenReview forum: "Gradient Descent Provably Solves Nonlinear Tomographic Reconstruction"
_ICLR.cc/2024/Conference — Submitted to ICLR 2024_

### Official Review · Reviewer_TLot · 2023-10-28

**Soundness:** 2 fair
**Presentation:** 4 excellent
**Contribution:** 2 fair
**Rating:** 5
**Confidence:** 4

**Summary:**

The authors consider the computed tomography (CT) reconstruction problem. Specifically, they consider a case with strong reflectors in the scene of interest (like a metal implant). When such materials are present, they argue that a preprocessing step that is normally used to obtain a linear inverse problem is not numerically stable. In contrast, they propose to solve the resulting non-convex problem directly through gradient descent. They show that, if the observation matrix were a sample from a Gaussian distribution, and if the number of measurements exceed a lower bound, with high probability, gradient descent converges to the true solution at a geometric rate.
They also extend the results to the case where the number of measurements are fewer but this reduction is compensated by a regularizer that captures the structure in the family of signals that the solution belongs to.
In an experiments section, they compare their proposed reconstructions against the linearized model (where the Fourier-slice theorem applies).

**Strengths:**

The paper is clear, well-written, and points to a relevant sub-problem in CT reconstruction. They show that a simple approach like gradient descent, targeting a relatively simple reconstruction formulation, can provably solve the original problem under certain constraints. They also extend their results to the case where the number of measurements are reduced at the expense of making certain assumptions about the signal of interest, and come up with clear statements that they can prove. Reducing the number of measurements in CT is always welcome as that reduces the amount of radiation the patient is subjected to.

**Weaknesses:**

The assumptions made by Theorems 1 and 2, which form the main contribution of the paper do not match well the CT measurement setup. Specifically, in CT, the beams are well-structured, and if we consider the image of a beam, it does not resemble at all a sample from a 2D Gaussian field. Therefore, it's not clear to me how much the results on provable convergence carry over. If the authors can show or argue that the local pseudo-convexity property (in some reasonable neighborhood of the solution) carries over to the deterministic concentrated-beam setting, the connection with CT would be stronger. Since the paper is not concerned with proposing a non-obvious algorithm but proving the usefulness of a rather well-known algorithm, this is a serious shortcoming.

Less importantly, the experiments are not entirely convincing -- I think reconstructions with the linear model could have been improved by including regularizers.

**Questions:**

Here are some comments/questions, some of which are minor.

1. Page 2, "We propose a Gaussian model of eqn 1.2" : Please clarify what you mean by a Gaussian model? I guess you're referring to modelling $a_i$ to be samples of a Gaussian field, but that's not clear at all at this stage.

2. Page 2, beginning of Section 2, "(1) we model $a_i$ as a standard Gaussian..." : This strongly contradicts the opening sentence of Section 2, where you mention "$a_i$ are sparse, nonnegative, highly structured, dependent on $i$". What is the motivation behind this modelling? Is it mainly to obtain provable results? I understand some deviation from an accurate model, but this is taking it a bit extreme.

3. The second sentence of Definition 2 ("Throughout...") doesn't look like it's part of the definition.

4. Page 5, footnote 1 : It's worth bringing in to the main text and clarifying what $m$ should exceed.

5. In general, I would welcome concrete typical values for the constants $c_i$ that appear in theorems, for specific cases/resolution.

6. Section 5.1 : Can you actually expect to see inside the metal? Here, I think the regularizer plays a more crucial role. What happens if you were to use a regularized formulation but with a linear measurement model (e.g., using one of the iterative solvers like FISTA)?

---

> ### Author Response · Authors · 2023-11-21
> **Response (part 1)**
>
> Thanks for your thoughtful comments on our work! We respond to individual comments and questions inline below.
>
> >The assumptions made by Theorems 1 and 2, which form the main contribution of the paper do not match well the CT measurement setup. Specifically, in CT, the beams are well-structured, and if we consider the image of a beam, it does not resemble at all a sample from a 2D Gaussian field. Therefore, it's not clear to me how much the results on provable convergence carry over. If the authors can show or argue that the local pseudo-convexity property (in some reasonable neighborhood of the solution) carries over to the deterministic concentrated-beam setting, the connection with CT would be stronger. Since the paper is not concerned with proposing a non-obvious algorithm but proving the usefulness of a rather well-known algorithm, this is a serious shortcoming.
>
> This is a great point that we have now addressed in the revised paper (and in this comment) in two ways.
>
> First, we have added an explanation of the logical steps we followed to reach our Gaussian theoretical model, and why each of these steps is well-supported by prior literature. In particular, our Gaussian model follows from two approximations: (1) A projection in real-space is equivalent to a slice through the origin in Fourier space, according to the Fourier slice theorem. Our first approximation involves sampling random Fourier coefficients rather than radial slices in Fourier space. This approximation is somewhat standard in the literature, for example in the seminal compressive sensing paper by Candes, Romberg, and Tao (https://arxiv.org/abs/math/0503066). (2) Our second approximation is to replace random Fourier coefficients with random Gaussian measurements. This approximation is also well supported by prior work, namely https://arxiv.org/abs/1506.03521 which shows that “structured random matrices behave similarly to random Gaussian matrices” in compressive sensing type settings.
>
> Second, and more importantly, we have independently verified that the key pseudoconvexity property we use in our proof—namely, the positivity of the correlation quantity <gradient of loss at z, z-x>/||z-x||^2, where x is the true signal and z is the current iterate—holds in practice even when we evaluate it using the true structure-preserving loss function instead of our Gaussian model. https://imgur.com/5rU3qgH (and the new Figure 1 in the revised manuscript) is a plot of the correlation value <gradient of loss at z, z-x>/||z-x||^2 where x is the synthetic phantom with ||x||= 7.4; the x-axis distance is normalized by this value so that the origin has normalized distance 1 relative to the global optimum. We sample 100 values of z at various distances from x, and show that this correlation is empirically positive for a wide range of distances including the distance between the origin and the true signal x. This experiment uses the true forward model that respects the ray structure, verifying that the same pseudoconvexity property we show in our Gaussian model also holds in the full forward model.
>
> >Less importantly, the experiments are not entirely convincing -- I think reconstructions with the linear model could have been improved by including regularizers.
>
> To clarify, the baseline linearized method in the synthetic experiment is including exactly the same regularization (by total variation) as the nonlinear method; we also note that this synthetic experiment is not in the underdetermined regime, as more measurements are provided than there are voxels in the reconstruction. The baseline linearized method in the real experiment is not using gradient descent, but rather using a state-of-the-art proprietary algorithm provided by our industrial collaborators. This real-data linearized baseline is the closest we have to a ground truth volume. In the revised paper, we have added an additional comparison to a more standard (and open-source) linearized baseline algorithm, FDK, which also exhibits worse metal artifacts than our nonlinear reconstruction. The updated figure is in the revised manuscript, and at https://imgur.com/a/PmtZz2O for convenience.

---

> > ### Author Response · Authors · 2023-11-21
> > **Response (part 2)**
> >
> > >Page 2, "We propose a Gaussian model of eqn 1.2" : Please clarify what you mean by a Gaussian model? I guess you're referring to modelling a_i  to be samples of a Gaussian field, but that's not clear at all at this stage.
> >
> > Yes, this is what we mean. We have clarified this in the revised paper.
> >
> > >Page 2, beginning of Section 2, "(1) we model a_i as a standard Gaussian..." : This strongly contradicts the opening sentence of Section 2, where you mention "a_i are sparse, nonnegative, highly structured, dependent on i". What is the motivation behind this modelling? Is it mainly to obtain provable results? I understand some deviation from an accurate model, but this is taking it a bit extreme.
> >
> > Yes, this is to make the analysis more tractable. Please refer to our response above regarding the connection between the theoretical model and the true/practical model.
> >
> > >The second sentence of Definition 2 ("Throughout...") doesn't look like it's part of the definition.
> >
> > Good point; that notation is only used in the footnote on page 5. In the revised version we have placed the definition next to the comment where it is used.
> >
> > >Page 5, footnote 1 : It's worth bringing in to the main text and clarifying what m should exceed.
> >
> > Thank you for the suggestion; it has been implemented in the revised paper.
> >
> > >In general, I would welcome concrete typical values for the constants c_i that appear in theorems, for specific cases/resolution.
> >
> > We expect the values to be of the order of 10s. However, we note that exact calculation of these constants is cumbersome and this type of proof is more about demonstration of the result holding with numerical constants rather than identifying the exact values, for which asymptotic techniques are often used instead.
> >
> > >Section 5.1 : Can you actually expect to see inside the metal? Here, I think the regularizer plays a more crucial role. What happens if you were to use a regularized formulation but with a linear measurement model (e.g., using one of the iterative solvers like FISTA)?
> >
> > The goal is less to see inside the metal (which at some point becomes unobservable, as the reviewer suggests), but to at least not have metal artifacts obscure tissue outside the metal. It is true that regularization can help reduce the severity of artifacts, but some noise amplified by the linearization step is unrecoverable even with regularization. For example, our synthetic experiments use the same regularization for both linearized and nonlinear reconstruction, and our real experiment uses a state-of-the-art linearized baseline that surely includes regularization (though the algorithm itself is proprietary). We have clarified the use of regularization in the linearized methods in the revised paper.
> >
> > Please let us know if we've addressed your concerns, or if there are any followup comments or questions. Thanks again for your time and thoughtful engagement with our work!

---

> > > ### Author Response · Authors · 2023-11-23
> > >
> > > As the deadline for author-reviewer interaction is closing tonight, we wanted to check in and ask if the reviewer has any remaining concerns that we may address. Thanks for your time and consideration of our work!

---

### Official Review · Reviewer_qFK5 · 2023-10-30

**Soundness:** 2 fair
**Presentation:** 2 fair
**Contribution:** 2 fair
**Rating:** 8
**Confidence:** 1

**Summary:**

Due to a medical emergency, I am unable to assess this article.

**Strengths:**

N/A

**Weaknesses:**

N/A

**Questions:**

N/A

---

> ### Author Response · Authors · 2023-11-21
>
> We wish you all the best in recovering from this medical emergency!

---

### Official Review · Reviewer_MBT4 · 2023-10-31

**Soundness:** 2 fair
**Presentation:** 2 fair
**Contribution:** 2 fair
**Rating:** 3
**Confidence:** 2

**Summary:**

The paper studies a nonlinear CT reconstruction model and provides global convergence for the proposed algorithms for unregularized and regularized problems. To justify the theoretical results, experiments are conducted on one synthetic data set and one real cone-beam CT data set. Some related works and further discussions are presented.

**Strengths:**

Detailed convergence analysis for the proposed algorithms would be the major strength. Besides, numerical experiments on CT data sets are conducted to justify the theoretical results.

**Weaknesses:**

1. The paper does not seem to use the standard ICLR paper template.
2. Regularization techniques have been widely used in solving inverse problems to address the ill-posedness. It is not clear about the contributions and novelty of the proposed approaches.
3. In the experiments, there are no other related works for comparison. So, it is hard to see whether the proposed performance is standing out.

**Questions:**

Why was the regularization term in the objective function in (2.3) recast as an inequality constraint in (4.1)? Will they give two different solutions? In addition, how was $x$ in (4.1) selected/constructed?

---

> ### Author Response · Authors · 2023-11-21
>
> Thanks for your time reviewing our paper! We respond to individual comments and questions inline below.
>
> >The paper does not seem to use the standard ICLR paper template.
>
> This is interesting–we are using the ICLR 2024 LaTeX template, but indeed it seems that the header line is missing in the original submission due to a package clash. Thanks for pointing this out; it has been resolved in the revised manuscript.
>
> >Regularization techniques have been widely used in solving inverse problems to address the ill-posedness. It is not clear about the contributions and novelty of the proposed approaches.
>
> The main problem we are addressing is separate from the ill-posedness problem from undersampling. Rather, we are addressing the issue of sensitivity to absorptive media like metal, which arises because of the poorly conditioned logarithmic preprocessing used to linearize the measurements in the standard reconstruction pipeline. This problem would persist with the linearized forward model even in the case of unlimited measurements. For example, our synthetic experiments are not in the ill-posed regime; instead they are overdetermined and have no added measurement noise, yet linearized reconstruction fails because the logarithmic preprocessing is sensitive even to the noise of machine precision.
>
> >In the experiments, there are no other related works for comparison. So, it is hard to see whether the proposed performance is standing out.
>
> We have added a standard linearized baseline method, FDK, in our real-data experiment, and we find that it also exhibits severe artifacts. The updated figure is in the revised manuscript, and at https://imgur.com/a/PmtZz2O for convenience. Please also note that the primary contribution of our work is the theoretical analysis, which is the first of its kind for this nonlinear forward model. The experiments are provided as a preliminary validation to support the theoretical results; they are not intended as a comprehensive comparison.
>
> >Why was the regularization term in the objective function in (2.3) recast as an inequality constraint in (4.1)? Will they give two different solutions?
>
> These two formulations (2.3 and 4.1) are equivalent in the sense that for any value of R(x) in the inequality constraint (4.1) there is a value of lambda in the regularization term (2.3) that will yield exactly the same solution. We also note that our theoretical results are robust to the exact choice of this regularization weight or inequality constraint threshold (see e.g. section 6.4 in https://arxiv.org/pdf/1507.04793.pdf), but we have stated the results in this form (with a precise choice of regularization or constraint) for simplicity of exposition.
>
> >In addition, how was x in (4.1) selected/constructed?
>
> In (4.1) the x in the constraint refers to the unknown ground truth signal. Even if the signal is unknown, this formulation assumes that some structure of x is known via the value of the function R(x). In practice this value R(x) can be viewed as a hyperparameter.
>
>
> Please let us know if we've addressed your concerns, or if there are any followup comments or questions. Thanks for your time and thoughtful engagement with our revision!

---

> > ### Author Response · Authors · 2023-11-23
> >
> > As the deadline for author-reviewer interaction is closing tonight, we wanted to check in and ask if the reviewer has any remaining concerns that we may address. Thanks for your time and consideration of our work!

---

### Official Review · Reviewer_1R2J · 2023-11-07

**Soundness:** 3 good
**Presentation:** 3 good
**Contribution:** 3 good
**Rating:** 6
**Confidence:** 4

**Summary:**

This article proposes a tomographic reconstruction method starting from the Beer-Lambert measurements on projections without linearisation. The resulting formulation is non-convex but the authors show its global optimum can be reached by gradient descent with high probability, with sufficient measurements, even in the presence of regularisation, which is the main result of the paper.

**Strengths:**

The paper is interesting and relatively easy to read. The approach is also fairly straightforward, although non-trivial, and it is surprising that is has not been attempted before. The mathematical and optimisation approach is well carried out.

**Weaknesses:**

Experiments are unconvincing, particularly Fig.2, where except around the high-density object (the crown), the quality of the linearised reconstruction is visibly better. Since the image is a phantom, a ground-truth should have been provided. The experiments on the Shepp-Logan phantom are too simple to be convincing.

Maybe this is due to a poor choice of hyper-parameter or a sub-optimal regulariser.

Reference code does not seem to be provided, limiting reproducibility.

**Questions:**

- How could you improve the fuzziness of Fig.2 ? This is essential because the current result does not sell your method well.
- There are many other non-linear aspect to real tomography, for example the phenomenon of beam hardening (BH), due to the fact that the X-ray source is not monochromatic and that lower-energy photons are absorbed more than higher-energy ones. Can you carry over some of your methodology to reduce the artefacts caused by BH?
- Could you compare with other non-linear tomography approaches.

---

> ### Author Response · Authors · 2023-11-21
>
> Thanks for your thoughtful comments on our work! We respond to individual comments and questions inline below.
>
> >Experiments are unconvincing, particularly Fig.2, where except around the high-density object (the crown), the quality of the linearised reconstruction is visibly better. Since the image is a phantom, a ground-truth should have been provided. The experiments on the Shepp-Logan phantom are too simple to be convincing.
>
> Just to clarify, when we describe the dental imaging target as a phantom, it is because it is not a living patient. This phantom is a real human skull, not a synthetic function, so we do not have a ground truth volume. We have removed the term “phantom” in the revised paper, when describing this skull dataset, to avoid any confusion for future readers. For this real dataset the linearized reconstruction is the closest we have to a ground truth; it is the state-of-the-art reconstruction provided by an industrial collaborator (using a proprietary method). We have also augmented our real-data experiment with a comparison to a more standard and open-source linearized baseline method, FDK, which likewise suffers metal artifacts as well as generally lower quality compared to either our nonlinear reconstruction or the commercial reference linearized reconstruction.
>
> >Maybe this is due to a poor choice of hyper-parameter or a sub-optimal regulariser.
>
> Indeed, by slightly reducing the total variation regularization we are able to reduce the blurriness in our nonlinear reconstruction. Thanks for the suggestion! The updated figure with our tuned results and the comparison to FDK is in the revised manuscript, and at https://imgur.com/a/PmtZz2O for convenience.
>
> >Reference code does not seem to be provided, limiting reproducibility.
>
> Our anonymized code has been uploaded as a supplemental file. We will also release the code publicly upon publication.
>
> >How could you improve the fuzziness of Fig.2 ? This is essential because the current result does not sell your method well.
>
> Please refer to our response above.
>
> >There are many other non-linear aspect to real tomography, for example the phenomenon of beam hardening (BH), due to the fact that the X-ray source is not monochromatic and that lower-energy photons are absorbed more than higher-energy ones. Can you carry over some of your methodology to reduce the artefacts caused by BH?
>
> This is a great suggestion that we will consider in followup work.
>
> >Could you compare with other non-linear tomography approaches.
>
> We agree it’s always preferable to compare to strong baselines. In this case, we are not aware of existing algorithms that use the nonlinear tomography forward model rather than the linearized version–though if the reviewer has suggestions we would be happy to look into them. To be clear, when we say non-linear we are referring to the measurement model, not the reconstruction algorithm. For example, there are some nonlinear methods (e.g. Neural Adaptive Tomography https://dl.acm.org/doi/abs/10.1145/3528223.3530121) that use a nonlinear model as the volume representation (e.g. neural net), but to our knowledge these still use the same linearized forward model and thus face the same sensitivity to absorptive media like metal.
>
> Please let us know if we've addressed your concerns, or if there are any followup comments or questions. Thanks again for your time and thoughtful engagement with our work!

---

> > ### Author Response · Authors · 2023-11-23
> >
> > As the deadline for author-reviewer interaction is closing tonight, we wanted to check in and ask if the reviewer has any remaining concerns that we may address. Thanks for your time and consideration of our work!

---

### Author Response · Authors · 2023-11-21
**Summary/Combined Response (Key Improvements in the Revision)**

Thanks to the reviewers for your time and thoughtful comments on our work. We have responded to individual comments and questions directly for each review, but we would like to highlight some of the main improvements we made in the revised manuscript based on your suggestions.
- We have included an experimental validation of our Gaussian theoretical model, showing that the local pseudoconvexity property that underlies our proof in the Gaussian setting also holds empirically using the full measurement model (including ray structure). We have also added an explanation of how we arrived at this Gaussian model, namely following two approximation steps that are both well-supported in the literature. This discussion and experimental validation is in a bolded paragraph titled “Connecting Theory to Practice” on pages 3 and 4 in the revised manuscript. For convenience, the new figure is also at https://imgur.com/5rU3qgH.
- We have clarified that our dental CT experiment is using real measurements of a human skull (not a synthetic function), and that the linearized baseline in that experiment is the reference volume provided by our collaborators’ proprietary reconstruction algorithm, the closest we have to a ground truth. We have augmented our results by adding a comparison with a more standard, open-source linearized reconstruction method, FDK, which exhibits clear artifacts.
- We have also improved the quality of our nonlinear reconstruction in the real-data experiment by further tuning (reducing) the value of our regularization parameter to reduce blurriness. The updated figure with our tuned results and the comparison to FDK is in the revised manuscript, and at https://imgur.com/a/PmtZz2O for convenience.
- We have included our anonymized code in the supplementary materials.

---

### Meta-Review · Area_Chair_5vRS · 2023-12-10

**Metareview:**

The paper presents a method for CT reconstruction. In its current form, the reviewers found the brief experiments not to be compelling and no thorough comparisons were attempted, and a shortcoming was that the technique isn’t that new, but the focus is more on showing an existing algorithm is useful for this problem. While the focus is likely of significant interest for CT, it’s not clear that there would be much appeal for the ideas at ICLR. Ultimately, the sentiment of the three actual reviews leaned towards rejection.

**Justification For Why Not Higher Score:**

The reviewer who gave an 8 should not have submitted anything since they had an emergency and did not actually read the paper by their own admission. The other reviewers seemed negative in their assessments and the paper itself does not seem a good fit for the conference in the topic and approach. Very little experimental effort basically limited to a toy problem.

**Justification For Why Not Lower Score:**

NA

---

### Decision · Program_Chairs · 2024-01-16

Reject